# LIBRA: LEVERAGING TEMPORAL IMAGES FOR BIOMEDICAL RADIOLOGY ANALYSIS

## ABSTRACT

Radiology report generation (RRG) is a challenging task, as it requires a thorough understanding of medical images, integration of multiple temporal inputs, and accurate report generation. Effective interpretation of medical images, such as chest X-rays (CXRs), demands sophisticated visual-language reasoning to map visual findings to structured reports. Recent studies have shown that multimodal large language models (MLLMs) can acquire multimodal capabilities by aligning with pre-trained vision encoders. However, current approaches predominantly focus on single-image analysis or utilise rule-based symbolic processing to handle multiple images, thereby overlooking the essential temporal information derived from comparing current images with prior ones. To overcome this critical limitation, we introduce Libra, a temporal-aware MLLM tailored for CXR report generation using temporal images. **Libra** integrates a radiology-specific image encoder with a MLLM and utilises a novel Temporal Alignment Connector to capture and synthesise temporal information of images across different time points with unprecedented precision. Extensive experiments show that Libra achieves new state-of-the-art performance among the same parameter scale MLLMs for RRG tasks on the MIMIC-CXR. Specifically, Libra improves the RadCliQ metric by 12.9% and makes substantial gains across all lexical metrics compared to previous models.

## 1 INTRODUCTION

Radiology reports are a key component of biomedical radiology analysis, designed to enhance clarity and efficiency in medical communication. These structured reports are typically divided into distinct sections, such as *Findings, Impression, Indication, Technique, Comparison*, and *History*(Ganeshan et al., 2018). Serving as the primary medium for radiologists to convey their findings and conclusions from imaging studies like chest X-rays (CXRs), radiology reports play a crucial role in guiding diagnostic and therapeutic decisions for various diseases(Najjar, 2023). However, generating these reports manually is a complex and time-consuming task. Automating radiology report generation (RRG) presents a valuable opportunity to boost radiologist productivity, improve communication, and alleviate burnout (Zhang et al., 2020b). Despite its potential, RRG is a challenging task due to the intricate nature of medical imaging and the need for precise and accurate documentation.

Although the advancements in paradigm models like LLaVA (Liu et al., 2023) and InstructBLIP (Dai et al., 2023), they still exhibit a significant performance gap when applied to RRG tasks. In particular, their effectiveness diminishes in biomedical contexts due to the substantial differences between biomedical and general image-text pairs (Tu et al., 2023; Saab et al., 2024). Consequently, these models often perform at a surface level, akin to a layperson-level understanding, when dealing with specialised medical imaging tasks. Recent advancements have explored continued pre-training of general-purpose foundation models for medical tasks, but these models still fall short in addressing the complexities of medical image analysis (Li et al., 2023a; Chaves et al., 2024; Park et al., 2024).

Harnessing the full potential of multimodal large language models (MLLMs) for understanding and reasoning over multimodal data in the biomedical domain poses a significant challenge, primarily due to the granularity and specificity required in medical image analysis (Wang et al., 2023a). Preliminary attempts to apply MLLMs to RRG have largely concentrated on generating reports from a single image, often neglecting the crucial temporal relationships between images taken at different time points (Zhang et al., 2024b). The MIMIC-CXR Database (Johnson et al., 2019a) reveals that

**67%** of patients underwent at least two studies at different time intervals, underscoring the necessity of incorporating temporal context in clinical practice. This highlights a critical gap in existing models, which are not equipped to handle the dynamic progression of medical conditions reflected in sequential imaging studies.

In standard clinical practice, radiologists rely heavily on comparing current imaging results with previous studies to assess temporal changes. However, many models inadvertently introduce spurious references to prior examinations that do not exist during inference, leading to inaccurate reports. Recent advancements[1], such as MedVersa (Zhou et al., 2024) and MAIRA-2 (Bannur et al., 2024), can process multiple images simultaneously. These models insert visual tokens from different images at specific points within the textual input, hinging on the LLM to interpret reference information. Additionally, LLaVA-Rad (Chaves et al., 2024) utilised GPT-4V (OpenAI et al., 2024) for radiology report extraction and augmentation, eliminating hallucinations associated with prior image references in the dataset. Despite these efforts, existing models still lack integrated mechanisms for perceiving temporal information within their architectures, which restricts the model's ability to achieve the temporal awareness required for comprehensive reporting. Meanwhile, existing MLLMs typically utilise visual embeddings from the last or penultimate layer of the image encoder (Chen et al., 2023a; Zhang et al., 2024a), which can only capture the global characteristics of the image. However, RRG tasks require not only identifying findings but also capturing subtle details such as severity, extent, and the progression of findings (Sloan et al., 2024). Such fine-grained details are difficult to represent using embeddings from any single layer of the image encoder. (Jiang et al., 2024). Therefore, for the high-granularity demands of RRG tasks, relying solely on representation from a single layer limits the quality of report generation. To tackle these limitations, we aim to enhance the temporal awareness of MLLMs for RRG by tackling two main challenges:

○ It is non-trivial to design effective structures in MLLMs to handle prior study citations across various time points in the RRG task.

○ The scarcity of effective connectors in MLLMs capable of handling the high-granularity requirements of downstream tasks.

To overcome these challenges, we propose **Libra** (**L**everaging Temporal **I**mages for **B**iomedical **R**adiology **A**nalysis), a novel framework designed to incorporate temporal change information into the RRG task. Libra employs a pre-trained visual transformer, RAD-DINO (Pérez-García et al., 2024), as the image encoder to generate robust image features, which are then refined using a new adapter specifically designed for the temporal awareness, before being fed into the medical large language model (LLM), Meditron (Chen et al., 2023b). Through a two-stage training strategy, Libra demonstrates the potential of MLLMs for specialised radiology applications. Our modular approach integrates state-of-the-art open-source pre-trained models for image and text while using a temporal-aware adapter to align these modalities with the text embedding space. Extensive experiments show that Libra substantially outperforms existing MLLMs in RRG tasks, demonstrating superior cross-modal understanding and reasoning capabilities. Specifically, our contributions include:

- We present **Libra**, the temporal-aware MLLM capable of capturing temporal changes and overcoming the challenge of handling prior study citations, setting a new state-of-the-art performance in RRG tasks on the MIMIC-CXR dataset among MLLMs of the same parameter scale.

- We designed the **Temporal Alignment Connector (TAC)** with two components: the Layerwise Feature Extractor (LFE), which extracts high-granularity image feature embedding from the encoder, and the Temporal Fusion Module (TFM), which integrates temporal references from prior studies. This enhances Libra's ability to capture and utilise temporal information in the RRG task.

## 2 LIBRA: BEING AWARE OF TEMPORAL CHANGES IN CHEST X-RAYS REPORT GENERATION

### 2.1 MODEL ARCHITECTURE

Libra model follows the common architecture of MLLMs, such as LLaVA, which includes an image encoder, a text decoder (i.e., an LLM pre-trained based on text-only data) and an adapter module for mapping visual representations into text space. In this work, we utilise a frozen biomedical image encoder, RAD-DINO (Pérez-García et al., 2024), a visual transformer extensively pre-trained on medical scans using the DINOv2 image-only self-supervised learning approach (Oquab et al., 2024).

---

[1]Detailed related work is discussed in Appx. A.1, and our research objectives are explained in Appx. A.2.

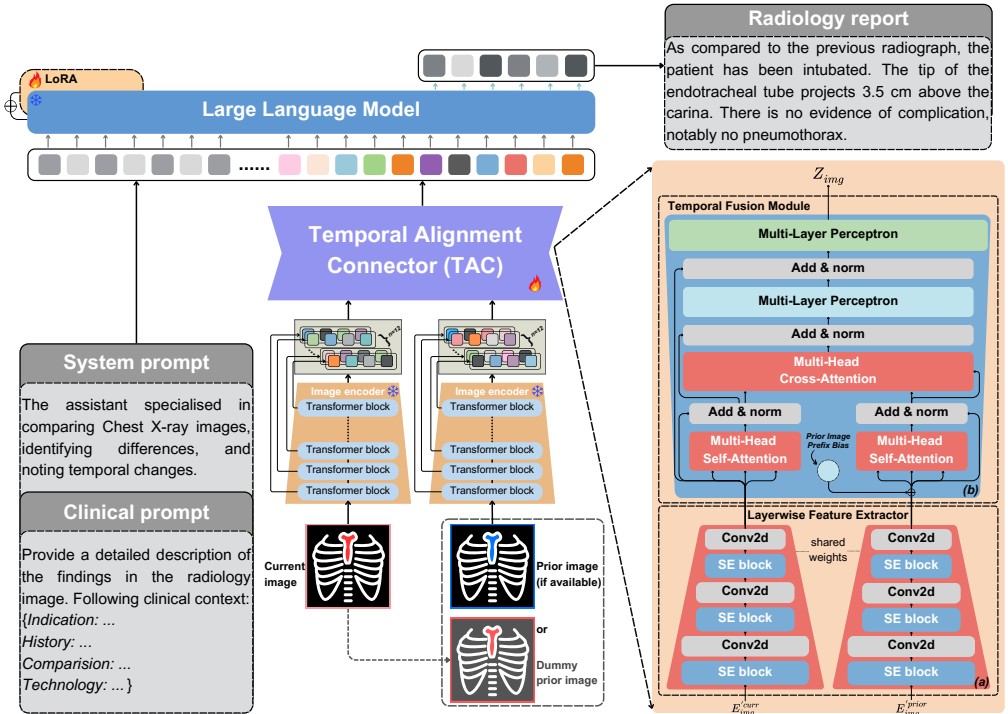

Figure 1: The overall architecture of Libra. The core component of the model is the Temporal Alignment Connector (TAC), designed to process a set of temporal images. It comprises two parts: *(a)* the Layerwise Feature Extractor (LFE), and *(b)* the Temporal Fusion Module (TFM).

The LLM is deployed by Meditron-7B (Chen et al., 2023b), which builds on Llama-2 (Touvron et al., 2023) and further pre-trained on a specialised medical corpus. However, for the adapter module, we specifically designed a Temporal Alignment Connector (TAC) to capture and integrate temporal information across two images from different time points, as shown in Figure 1. The TAC bridges the image encoder and LLM. It consists of two components: the Layerwise Feature Extractor (LFE), which extracts high-granularity image representations, and the Temporal Fusion Module (TFM), which integrates temporal references from prior studies. This design enables Libra to effectively manage temporal data and enhances its ability to generate accurate and coherent radiology reports.

## 2.2 TEMPORAL ALIGNMENT CONNECTOR (TAC): NEW ADAPTER DESIGN

To address the challenges of integrating temporal information and aligning high-granularity image features in RRG tasks, we developed the TAC module. It bridges the gap between visual features extracted from multiple temporal snapshots of a patient's CXRs and the language model. The TAC consists of two sub-modules: the LFE, which is to extract the layerwise representation of images, and the TFM, which is to capture the temporal-aware representation between two images.

### 2.2.1 LAYERWISE FEATURE EXTRACTOR (LFE)

To leverage abundant feature information in a pre-trained image encoder, we extract all image patch token features from each hidden layer for a given input image. Specifically, RAD-DINO has 12 hidden layers and processes $518 \times 518$ images into $14 \times 14$ patches, generating 1,369 patch token sequences per hidden layer for each image. We do not use the $[CLS]$ token. Following this, we obtain patch embeddings of the same dimension from each layer, denoted as $E^{\mathrm{img}} \in \mathbb{R}^{N \times D_{\mathrm{img}}}$, where $N$ is the number of patch tokens and $D_{\mathrm{img}}$ is the embedding dimension of the image encoder. Then, we combine them as $E'_{\mathrm{img}} = \{E_i^{\mathrm{img}}\}_{i=1}^{n}$, where $n$ is the number of hidden layers. Drawing from VGG (Simonyan & Zisserman, 2015), we employ a progressive compression strategy to reduce the layer dimension, ensuring that the LFE captures the most relevant features. Initially, we utilise Squeeze-and-Excitation (SE) Networks (Hu et al., 2019), which construct informative features by integrating both spatial and channel-wise information within local receptive fields at each layer.

The SE block is applied to obtain calibrated feature representations, using GELU (Hendrycks & Gimpel, 2023) as the activation function. Next, we employ a specialised pointwise convolution module to align the feature spaces across different layers, using a depthwise 2D convolution with filters and stride of 1, without bias. The dimensions are gradually reduced to better align with the text in the LLM. The compressed features are represented as $A_{\text{img}} \sim Conv2d_j^k(SE_j^k(E'_{\text{img}}))$, where $k$ is the original layer number and $j$ is the layer number after compression. Following the size-reduction pattern of convolutional layers in VGG, the image representations are compressed according to $\{k, j\} \in \{12, 6, 3, 1\}^2$. Through three stages of progressive compression, we obtain the final representation of image patch features. We define the function $LFE(\cdot)$ to project image features into the same dimension:

$$A_{\text{img}} = LFE(E'_{\text{img}}) \iff A_{\text{img}} = Conv2d_1^3(SE_1^3(A'')) \in \mathbb{R}^{1 \times N \times D_{\text{img}}}, \tag{1}$$

where $A'' = Conv2d_3^6(SE_3^6(A'))$, $A' = Conv2d_6^{12}(SE_6^{12}(E'_{\text{img}}))$. The LFE independently processes each image to produce a unified representation for temporal alignment, as **(a)** in Figure 1.

### 2.2.2 TEMPORAL FUSION MODULE (TFM)

The fusion module is inspired by the transformer decoder. It utilises prior images as auxiliary information to weight the current image, generating a refined, temporal-aware representation suitable for the LLM's latent space. The TFM uses an attention mechanism to dynamically learn relationships between image pairs, enabling the model to adapt to various temporal changes.

**Prior Image Prefix Bias:** The dataset contains samples with and without a prior image. The patch token features of the current and previous images processed by the LFE are denoted as $A_{\text{img}}^{\text{curr}}$ and $A_{\text{img}}^{\text{prior}}$. If no true prior image exists, we set $A_{\text{img}}^{\text{prior}} = A_{\text{img}}^{\text{curr}}$, effectively using the current image as a dummy prior. To differentiate this case, we introduce a trainable bias $b_{\text{prior}}$. Following the attention mechanism's scaling techniques for adjusting hidden space degrees of freedom with a chi-square distribution (Vaswani, 2017; Wang et al., 2018a), we apply an exponent of $\sqrt[4]{d}$ to the cosine similarity, where $d$ is the hidden dimension of the LLM, to obtain a weighted bias addition to $A_{\text{img}}^{\text{prior}}$:

$$A_{\text{img}}'^{\text{prior}} = A_{\text{img}}^{\text{prior}} + b_{\text{prior}} \cdot \left( \frac{Cos(A_{\text{img}}^{\text{curr}}, A_{\text{img}}^{\text{prior}}) + 1}{2} \right)^{\sqrt[4]{d}}, \tag{2}$$

This nonlinear scaling amplifies higher similarity values, effectively modulating the influence of prior image features. When no true prior image is available, the high similarity score ensures that the effect of the dummy prior is adequately represented. This adjustment prevents samples with a dummy prior image from undergoing redundant rounds of parallel multi-head self-attention during subsequent propagation through the transformer blocks, as shown in Figure 1.

**Transformer Block:** Our transformer block includes a multi-head cross-attention sub-layer, multi-head self-attention sub-layers, and two multi-layer perceptron (MLP) sub-layers. As shown in Figure 1 **(b)**. The paired $(A_{\text{img}}^{\text{curr}}, A_{\text{img}}'^{\text{prior}})$ are processed with layer normalization and residual connections:

$$T_{\text{curr}}^{\text{self}} = LayerNorm(A_{\text{img}}^{\text{curr}} + SelfAttn(A_{\text{img}}^{\text{curr}}; A_{\text{img}}^{\text{curr}})), \tag{3}$$

$$T_{\text{prior}}^{\text{self}} = LayerNorm(A_{\text{img}}'^{\text{prior}} + SelfAttn(A_{\text{img}}'^{\text{prior}}; A_{\text{img}}'^{\text{prior}})), \tag{4}$$

$$T_{\text{img}}^{\text{cros}} = LayerNorm(T_{\text{curr}}^{\text{self}} + CorssAttn(T_{\text{curr}}^{\text{self}}; T_{\text{prior}}^{\text{self}})), \tag{5}$$

$$T_{\text{img}}^{\text{out}} = LayerNorm(A_{\text{img}}^{\text{curr}} + MLP_{\text{attn}}(T_{\text{img}}^{\text{cros}})), \tag{6}$$

where $MLP_{\text{attn}}$ is a simple neural network composed of two fully connected layers with GELU as the activation function. After that, the features are processed through $MLP_{\text{final}}$, a straightforward neural network consisting of four fully connected layers with the same activation function, but with hidden dimensions matching those of the LLM. We define our fusion module function as $TFM(\cdot)$.

$$Z_{\text{img}} = TFM(A_{\text{img}}^{\text{curr}}, A_{\text{img}}^{\text{prior}}) \iff Z_{\text{img}} = MLP_{\text{final}}(T_{\text{img}}^{\text{out}}), \tag{7}$$

Subsequently, $Z_{\text{img}} \in \mathbb{R}^{N \times d}$ becomes the input sequence that LLM can comprehend, where $N$ is the number of patch tokens and $d$ is the hidden dimension of LLM. The final output is a refined feature representation that encapsulates the temporal evolution of the patient's condition, making it suitable for the language model to generate accurate and contextually aware radiology reports.

---

[2]Since RAD-DINO has 12 hidden layers, the prime factorisation chain provides the factors as $\{12, 6, 3, 1\}$.

## 2.3 PROMPT DESIGN

To enhance Libra's ability to perceive temporal changes and integrate medical information in the RRG task, we designed a comprehensive prompting strategy comprising a system prompt and detailed clinical prompts, as shown in Figure 1. The system prompt is crafted to enable the LLM to recognise temporal variations: "The assistant specialised in comparing Chest X-ray images, identifying differences, and noting temporal changes." Following this, additional sections of the report, such as {*Indication*}, {*History*}, {*Comparison*}, and {*Technique*}, are incorporated into the clinical instructions (e.g., in Appx. B.4). The clinical prompt to Libra is: "Provide a detailed description of the findings in the radiology image. Following clinical context: {...}." If no clinical instructions are available, the prompt defaults to: "Provide a detailed description of the findings in the radiology image." After tokenizing and embedding the prompt and the answer, we insert the image patch tokens at the specified location, typically between the system prompt and the clinical prompts.

## 2.4 TEMPORAL-AWARE TRAINING

In this study, we focus on frontal-view images, either posterior-anterior (PA) or anterior-posterior (AP), and the *Findings* sections of radiology reports, as they contain the most direct clinical observations. We adopt a two-stage training strategy inspired by recent advancements in MLLM fine-tuning techniques (McKinzie et al., 2024). This proposed approach aims to progressively teach the model essential skills, including visual feature alignment and temporal information extraction. Libra's training process involves two stages: temporal feature alignment and downstream task fine-tuning.

In the first stage, the visual encoder and LLM weights are frozen, and the TAC is trained with *Findings* and *Impression* generation, along with CXR-related VQA tasks to extract high-quality image representations and capture temporal changes. In the second stage, we apply Low-Rank Adaptation (LoRA) (Hu et al., 2021) to fine-tune the pre-trained LLM on the *Findings* section generation task, keeping the visual encoder and connector weights frozen. Unlike traditional full fine-tuning, LoRA achieves comparable performance with significantly lower training costs. The detailed training configuration, including learning rate schedules and model parameters, is provided in Appx. B.1.

## 3 EXPERIMENTS

### 3.1 TASK AND DATASET

**Task Description** We focus on generating the *Findings* section of radiology reports for a set of frontal CXRs to ensure fair comparison with previous work. The *Findings* section contains descriptions by radiologists of both normal and abnormal findings. Typically, these CXRs are accompanied by additional sections such as *Indication* and *Technique*, which provide the rationale for the study, including clinical history or specific requests from the referring physician. While not directly related to diagnostic interpretation, these sections serve as routine records and assist the model in better understanding temporal changes between images across different periods. Therefore, we incorporate clinical instructions about the current image as prompts to guide Libra to complete the RRG task.

The most common CXR is frontal views, either PA or AP. While other views, such as lateral images, are occasionally used for diagnostic purposes, they mainly serve as supplementary tools to aid in interpreting anatomical structures more comprehensively (Islam et al., 2023). Therefore, the frontal view remains the standard perspective for clinical interpretation of CXRs, and our study is consistent with prior work in RRG tasks, including studies by Chaves et al. (2024) and Hyland et al. (2024). For both the current and prior images, we exclusively use a single frontal view.

**Dataset Description** To train Libra, we utilise the MIMIC-CXR dataset (Johnson et al., 2019b) and its derivative datasets, Medical-Diff-VQA (Hu et al., 2023) and MIMIC-Ext-*MIMIC-CXR-VQA* (Bae et al., 2023), as shown in the Table 1. All datasets are split according to the official labels to ensure no data leakage. Detailed descriptions of datasets can be found in the Appx. B.2.

Inspired by the multi-stage training techniques commonly employed in recent MLLMs fine-tuning methods (Zhu et al., 2023), we adopt a two-stage training strategy to infuse multimodal capabilities into the model. In the first stage, we leverage a large-scale dataset to train the adapter, primarily focusing on learning the distribution of CXR token embeddings and the general relationships between

Table 1: Datasets used for training and evaluating Libra include statistics on the proportion of samples that contain prior images. In the first stage of training, we utilise the entire dataset, while the second stage focuses on fine-tuning the model on downstream tasks.

| Data source | Task Type | # Samples | | | % Has Prior | | |
|---|---|---|---|---|---|---|---|
| | | Train (%) | Valid (%) | Test (%) | Train | Valid | Test |
| MIMIC-CXR | Findings | 162 955 (13.43%) | 1286 (0.88%) | 2461 (2.78%) | 58.43 | 60.11 | 86.03 |
| | Impression | 199 548 (16.45%) | 1671 (1.14%) | 2343 (2.64%) | 64.85 | 67.09 | 85.49 |
| Medical-Diff-VQA | Difference | 131 563 (10.85%) | 16 372 (11.17%) | 16 389 (18.48%) | 100 | 100 | 100 |
| | Abnormality | 116 394 (9.59%) | 14 512 (9.90%) | 14 515 (16.37%) | 100 | 100 | 100 |
| | Presence | 124 654 (10.28%) | 15 549 (10.61%) | 15 523 (17.51%) | 100 | 100 | 100 |
| | View | 44 970 (3.71%) | 5696 (3.89%) | 5599 (6.31%) | 100 | 100 | 100 |
| | Location | 67 187 (5.54%) | 8510 (5.81%) | 8496 (9.58%) | 100 | 100 | 100 |
| | Level | 53 728 (4.43%) | 6722 (4.59%) | 6846 (7.72%) | 100 | 100 | 100 |
| | Type | 22 067 (1.82%) | 2709 (1.85%) | 2702 (3.05%) | 100 | 100 | 100 |
| MIMIC-Ext-*MIMIC-CXR-VQA* | Presence | 109 455 (9.02%) | 26 153 (17.84%) | 4566 (5.15%) | 0 | 0 | 0 |
| | Anatomy | 37 952 (3.13%) | 10 210 (6.96%) | 1963 (2.21%) | 0 | 0 | 0 |
| | Attribute | 49 948 (4.12%) | 13 111 (8.94%) | 2578 (2.91%) | 0 | 0 | 0 |
| | Abnormality | 60 692 (5.00%) | 16 109 (10.99%) | 3199 (3.61%) | 0 | 0 | 0 |
| | Size | 16 000 (1.32%) | 4000 (2.73%) | 705 (0.80%) | 0 | 0 | 0 |
| | Plane | 7992 (0.66%) | 1992 (1.36%) | 386 (0.44%) | 0 | 0 | 0 |
| | Gender | 7992 (0.66%) | 1992 (1.36%) | 396 (0.45%) | 0 | 0 | 0 |
| **Total** | Multi-type | 1 213 097 (100%) | 146 594 (100%) | 88 669 (100%) | 64.73 | 49.09 | 83.67 |

CXR and text tokens across the MIMIC dataset. The second stage fine-tuning the model on downstream tasks, where the text decoder (i.e., the LLM) is refined to focus on aligning high-granularity information between CXRs and the *Findings* section of radiology reports.

In addition to *Findings* section generation, the temporal feature alignment stage also uses the *Impression* section and VQA tasks. The *Impression* section summarises diagnoses and proposes further investigations (Babar et al., 2021), aiding the alignment between CXRs and their textual descriptions. We use the same system and clinical prompts as for *Findings*, replacing 'Findings' with 'Impression'. For VQA, system prompts remain the same, while clinical prompts adapt to medical-specific questions, guiding the model to generate corresponding captions. These fine-grained VQA tasks expand the MLLM's biomedical vocabulary and improve the alignment of image-text annotations.

### 3.2 Evaluation Metrics

We evaluate the generated reports using lexical and radiology-specific metrics, following established protocols.. For lexical metrics, we use ROUGE-L (Lin, 2004), BLEU-$\{1, 4\}$ (Papineni et al., 2002), METEOR (Banerjee & Lavie, 2005) and BERT (Devlin et al., 2019). For radiology-specific metrics, we use RadGraph-F1 (Jain et al., 2021), $RG_{ER}$ (Delbrouck et al., 2022a), F1-CheXpert (Irvin et al., 2019a), CheXbert vector similarity (Yu et al., 2022), as well as RadCliQ version 0 Yu et al. (2022).

Additionally, we introduce a novel temporal entity F1 score (called $F1_{temp}$) to assess the model's ability to capture temporal information. Clinical metrics typically emphasise the accuracy of medical findings, prioritising the detection of clinically relevant entities. In contrast, the temporal entity score specifically measures the accuracy of entities related to progression over time described in the report. Detailed descriptions of all metrics and an illustrative $F1_{temp}$ analysis are provided in Appx. B.3.

**Temporal Entity F1** Following the work of Bannur et al. (2023), we set a reward list that includes common radiology-related keywords associated with temporal changes. Temporal entities are then extracted from both the ground truth ($E_{gt}$) and the generated reports ($E_{gr}$). Importantly, no stemming or lemmatization is applied during token processing to preserve the precise description of temporal changes. After extraction, we compute precision ($P_{temp}$) and recall ($R_{temp}$), which are subsequently used to calculate the temporal entity score, defined as the harmonic mean of precision and recall (Van Rijsbergen, 1974), also known as the F1 score.

$$P_{temp} = \frac{|E_{gr} \cap E_{gt}| + \epsilon}{|E_{gt}| + \epsilon}; \quad R_{temp} = \frac{|E_{gr} \cap E_{gt}| + \epsilon}{|E_{gr}| + \epsilon}; \quad F1_{temp} = (1+\beta^2)\cdot\frac{P_{temp}R_{temp}}{\beta^2 \cdot P_{temp} + R_{temp}} \quad (8)$$

where $\epsilon$ is a small value, set to a default of $1 \times 10^{-10}$, to prevent division by zero (it is also added to the numerator for special cases where no temporal entities are present in the ground truth). The coefficient $\beta$ controls the balance between precision and recall, with a default value of 1.

## 3.3 MAIN RESULT

Although MIMIC-CXR provides an "official" test split, strict comparisons with prior studies remain challenging due to differences in test set inclusion criteria and pre-processing steps. For instance, Yu et al. (2022) and Jeong et al. (2023) included only one image per study, resulting in a test set of 1,597 samples, while Tanida et al. (2023) followed the Chest ImaGenome split (Wu et al., 2021b). Such variations in test set distributions can significantly impact the reported results (Park et al., 2024). To ensure fairness, we use a widely adopted test set focused on frontal-view CXRs, aligned with previous studies such as MAIRA-1 (Hyland et al., 2024) and LLaVA-Rad (Chaves et al., 2024) [3].

Additionally, the latest concurrent work M4CXR (Park et al., 2024) employs multi-turn chain-of-thought (Wei et al., 2023) prompting to generate reports, which differs from our task setup. Furthermore, we do not compare with the recent work MAIRA-2, as it is a MLLM specifically designed for grounded radiology report generation task, which incorporates lateral views and prior study reports for each subject within the input prompt. Bannur et al. (2024) emphasises a positive transfer between this task and general report generation, which is also beyond the scope of our study [4].

Taking these considerations into account, we compared our model with state-of-the-art models, including LLaVA-Med (Li et al., 2023a), CheXagent (Chen et al., 2024), GPT-4V (OpenAI et al., 2024), Med-PaLM (Tu et al., 2023), LLaVA-Rad and MAIRA-1. The results are shown in Table 2. Since many of these models are not publicly available, we present their evaluation results as reported in the original sources.

Table 2: Findings generation performance on the official MIMIC-CXR test split. ‡ denotes numbers provided by Chaves et al. (2024), and § represents numbers obtained from Hyland et al. (2024) The best performances in **bold**, and the second-best scores are underlined. '↓' indicates that lower is better. The percentage (%) indicates the improvement over the best existing model.

| Metric | LLaVA-Med[§] | CheXagent[‡] | GPT-4V[‡] | Med-PaLM | LLaVA-Rad | MAIRA-1 | Libra(%) |
|---|---|---|---|---|---|---|---|
| **Lexical:** | | | | | | | |
| ROUGE-L | 27.6 | 21.5 | 13.2 | 27.5 | 30.6 | 28.9 | **36.7** (19.9%) |
| BLEU-1 | 35.4 | 16.9 | 16.4 | 32.3 | 38.1 | 39.2 | **51.3** (30.9%) |
| BLEU-4 | 14.9 | 4.7 | 17.8 | 11.5 | 15.4 | 14.2 | **24.5** (37.6%) |
| METEOR | 35.3 | – | – | – | – | 33.3 | **48.9** (38.5%) |
| **Clinical:** | | | | | | | |
| RadGraph-F1 | 19.1 | – | – | 26.7 | – | 24.3 | **32.9** (23.2%) |
| RG$_{ER}$ | 23.8 | 20.5 | 13.2 | – | 29.4 | 29.6 | **37.6** (27.0%) |
| RadCliQ$_0$(↓) | 3.3 | – | – | – | – | 3.1 | **2.7** (12.9%) |
| CheXbert vector | 36.9 | – | – | – | – | 44.0 | **46.9** (6.59%) |
| *CheXpert-F1:* | | | | | | | |
| Micro-F1-14 | 42.7 | 39.3 | 35.5 | 53.6 | **57.3** | 55.7 | 55.9 (-2.4%) |
| Macro-F1-14 | 26.9 | 24.7 | 20.4 | 39.8 | 39.5 | 38.6 | **40.4** (1.5%) |
| Micro-F1-5 | 43.9 | 41.2 | 25.8 | 57.9 | 57.4 | 56.0 | **60.1** (3.8%) |
| Macro-F1-5 | 36.3 | 34.5 | 19.6 | 51.6 | 47.7 | 47.7 | **53.8** (4.3%) |

From Table 2 we can see that, for all traditional lexical metrics, Libra substantially outperforms the baseline models, achieving scores of 36.7 for ROUGE-L, 51.3 and 24.5 for BLEU-1 and BLEU-4 respectively, and 48.9 for METEOR. In clinical metrics, Libra also showed outstanding performance, substantially improving in RadGraph-F1 and its variant RG$_{ER}$, with scores of 32.9 and 37.6, respectively. Notably, Libra achieved the highest score (i.e. 2.7) on the radiologist-aligned RadCliQ metric. For CheXbert-derived metrics, it also secured the top position with a vector embedding matrix score of 46.9. In the CheXpert 5-class subset, Libra excelled with Micro-F1 and Macro-F1 scores of 60.1 and 53.8, respectively. In the CheXpert 14-class subset, Libra achieved the highest Macro-F1 score of 40.4, while its Micro-F1 score of 55.9 was only slightly below the leading model.

In summary, Libra demonstrated outstanding performance across both sets of metrics, with only minor gaps in one specific measure. The remarkable capabilities of Libra can be attributed to its innovative Temporal Alignment Connector which effectively captures temporal changes in the dataset, enabling the generation of highly accurate and clinically relevant reports.

---

[3]The test set consists of 2,461 frontal-view image-report samples.

[4]For comparisons with the latest concurrent and non-LLM-based models can be found in Appx. D.

## 4 ABLATION STUDY

To thoroughly assess the impact of various optimisation components, we conducted a series of ablation experiments, evaluating different modules and dataset expansions within Libra. All ablation models were tested on the MIMIC-CXR official test split for the *Findings* section generation task, using identical hyperparameter settings during training and inference.

**Q1: Does incorporating temporal information bring positive effects to Libra in RRG tasks?**

Temporal information is embedded in paired images and referenced in the corresponding radiology reports, reflecting changes over time through references to previous symptoms and descriptions of their progression. As indicated in Table 1, 86% of the test data includes true prior images. To assess whether Libra can effectively perceive and utilise temporal information during inference, we conducted separate evaluations using either true prior images or only dummy prior images. When true prior images were unavailable, the model treated the current image as a dummy prior image. As shown in Table 3, Libra demonstrated substantial enhancements across all metrics when true prior images were used as references. Specifically, the improvement in clinical scores exceeds that of lexical scores, indicating that temporal information is crucial for generating high-quality medical reports, beyond merely improving linguistic fluency. Notably, the $F1_{temp}$ score shows the most significant impact, with a difference of **7.41%**. This highlights Libra's capability to effectively leverage temporal changes provided by true prior images, thereby enhancing the quality of the generated *Findings* section and overall performance in the RRG task.

Table 3: Ablation results for Libra using true prior images or only dummy prior images. Values in (%) indicate the percentage improvement.

| Metric | Libra | |
|---|---|---|
| | w/ true | w/ dummy(%) |
| **Lexical:** | | |
| ROUGE-L | 36.66 | 36.17 (-1.34%) |
| BLEU-1 | 51.25 | 51.20 (-0.10%) |
| BLEU-4 | 24.54 | 24.33 (-0.86%) |
| METEOR | 48.90 | 48.69 (-0.43%) |
| BERTScore | 62.50 | 61.94 (-0.90%) |
| $F1_{temp}$ | 35.34 | 32.72 (-7.41%) |
| **Clinical:** | | |
| RadGraph-F1 | 32.87 | 32.42 (-1.37%) |
| $RG_{ER}$ | 37.57 | 36.92 (-1.73%) |
| $RadCliQ_0(\downarrow)$ | 2.72 | 2.76 (-1.47%) |
| CheXbert vector | 46.85 | 46.31 (-1.15%) |
| *CheXpert-F1:* | | |
| Micro-F1-14 | 55.87 | 55.25 (-1.11%) |
| Macro-F1-14 | 40.38 | 40.15 (-0.57%) |
| Micro-F1-5 | 60.07 | 58.93 (-1.90%) |
| Macro-F1-5 | 53.75 | 52.61 (-2.12%) |

**Q2: Does the Temporal Alignment Connector effectively improve model performance?**

Table 4: Results of ablation experiments for the Temporal Alignment Connector. '↓' indicates that lower is better. Values in (%) indicate the percentage decrease compared with the Libra-1.

| Metric | Libra-1 | w/o TFM | w/o LFE | w/o PIPB | w/o TAC |
|---|---|---|---|---|---|
| **Lexical:** | | | | | |
| ROUGE-L | 27.56 | 27.33 (-0.85%) | 27.21 (-1.27%) | 27.43 (-0.48%) | 26.17 (-5.04%) |
| BLEU-1 | 34.84 | 34.17 (-1.92%) | 34.21 (-1.82%) | 34.60 (-0.67%) | 33.03 (-5.20%) |
| BLEU-4 | 11.51 | 11.13 (-3.33%) | 11.11 (-3.47%) | 11.43 (-0.73%) | 10.02 (-12.98%) |
| METEOR | 35.50 | 35.06 (-1.24%) | 34.96 (-1.52%) | 35.28 (-0.62%) | 33.98 (-4.28%) |
| BERTScore | 55.87 | 55.60 (-0.49%) | 55.49 (-0.69%) | 55.74 (-0.23%) | 54.63 (-2.22%) |
| $F1_{temp}$ | 26.63 | 25.96 (-2.51%) | 26.21 (-1.57%) | 26.58 (-0.18%) | 25.39 (-4.65%) |
| **Clinical:** | | | | | |
| RadGraph-F1 | 22.52 | 22.20 (-1.42%) | 22.03 (-2.19%) | 22.35 (-0.74%) | 21.51 (-4.48%) |
| $RG_{ER}$ | 27.32 | 26.89 (-1.59%) | 26.72 (-2.19%) | 27.09 (-0.84%) | 25.97 (-4.96%) |
| $RadCliQ_0\ (\downarrow)$ | 3.10 | 3.12 (-0.65%) | 3.12 (-0.65%) | 3.11 (-0.32%) | 3.15 (-1.61%) |
| CheXbert vector | 42.02 | 41.57 (-1.07%) | 41.37 (-1.54%) | 41.92 (-0.24%) | 40.93 (-2.59%) |
| *CheXpert-F1:* | | | | | |
| Micro-F1-14 | 52.48 | 51.74 (-1.42%) | 51.68 (-1.53%) | 52.13 (-0.67%) | 51.13 (-2.57%) |
| Macro-F1-14 | 36.87 | 36.04 (-2.25%) | 36.12 (-2.03%) | 36.14 (-1.97%) | 35.85 (-2.76%) |
| Micro-F1-5 | 56.63 | 55.37 (-2.23%) | 55.79 (-1.49%) | 55.87 (-1.34%) | 54.51 (-3.74%) |
| Macro-F1-5 | 49.33 | 47.76 (-3.18%) | 47.82 (-3.06%) | 47.98 (-2.75%) | 47.22 (-4.28%) |

To evaluate the impact of TAC on Libra's performance in the RRG task, we initialised a model with the RAD-DINO (Pérez-García et al., 2024) image encoder, TAC, and Meditron (Chen et al., 2023b) as the LLM. A baseline experiment (Libra-1) was conducted by fine-tuning only the TAC for the *Findings* generation task. As shown in Table 4, we performed ablation studies by progressively removing different TAC components, including TFM, LFE, the Prior Image Prefix Bias (PIPB), and the entire TAC. Removing TFM restricted the model to processing only the current image, using a configuration similar to LLaVA (Liu et al., 2023) but with a four-layer MLP to align the image representation with the LLM's hidden dimensions. Without LFE, the model follows the LLaVA setup,

utilising the penultimate layer of the image encoder. The results indicate that removing any TAC submodule leads to a decline in all metrics compared to Libra-1. Removing TFM caused a notable drop in the $F1_{temp}$ score ($\downarrow > 2\%$), emphasising its role in capturing temporal information. Removing LFE especially decreases RadGraph-related scores, highlighting its importance in extracting detailed image features. The removal of PIPB impacted clinical scores more than lexical scores. Finally, removing the entire TAC results in substantial declines across all metrics, underscoring its critical role in integrating image details and temporal information[5].

**Q3: Are additional *Impression* and VQA datasets needed during the feature alignment?**

Table 5: Ablation experiments of dataset expansion in Libra. $\triangle$ indicates better performance at the same stage, while $\blacktriangledown$ indicates the opposite.

| Metric | Stage: 1 | | Stage: 2 | |
|---|---|---|---|---|
| | **Libra** | *Libra-f* | **Libra** | *Libra-f* |
| **Lexical:** | | | | |
| ROUGE-L | 27.27 | 27.56$^\triangle$ | 36.66 | 35.31$^\blacktriangledown$ |
| BLEU-1 | 41.24 | 34.84$^\blacktriangledown$ | 51.25 | 49.92$^\blacktriangledown$ |
| BLEU-4 | 13.59 | 11.51$^\blacktriangledown$ | 24.54 | 23.05$^\blacktriangledown$ |
| METEOR | 39.44 | 35.50$^\blacktriangledown$ | 48.90 | 47.99$^\blacktriangledown$ |
| BERTScore | 56.00 | 55.87$^\blacktriangledown$ | 62.50 | 61.28$^\blacktriangledown$ |
| $F1_{temp}$ | 24.80 | 26.63$^\triangle$ | 35.34 | 33.52$^\blacktriangledown$ |
| **Clinical:** | | | | |
| RadGraph-F1 | 20.45 | 22.52$^\triangle$ | 32.87 | 30.77$^\blacktriangledown$ |
| $RG_{ER}$ | 25.19 | 27.32$^\triangle$ | 37.27 | 35.44$^\blacktriangledown$ |
| $RadCliQ_0$ ($\downarrow$) | 3.31 | 3.10$^\triangle$ | 2.72 | 2.83$^\blacktriangledown$ |
| CheXbert vector | 35.33 | 42.02$^\triangle$ | 46.85 | 45.32$^\blacktriangledown$ |
| *CheXpert-F1:* | | | | |
| Micro-F1-14 | 43.63 | 52.48$^\triangle$ | 55.87 | 54.11$^\blacktriangledown$ |
| Macro-F1-14 | 25.68 | 36.87$^\triangle$ | 40.38 | 37.16$^\blacktriangledown$ |
| Micro-F1-5 | 49.75 | 56.63$^\triangle$ | 60.07 | 58.76$^\blacktriangledown$ |
| Macro-F1-5 | 40.40 | 49.33$^\triangle$ | 53.75 | 51.99$^\blacktriangledown$ |

To evaluate the impact of dataset expansion during the first stage on Libra's performance, we trained a model (Libra-f) using only the *Findings* data in the first stage, while Libra incorporated additional *Impression* and VQA data for alignment, in Table 5. After the first stage, Libra showed superior performance in lexical metrics compared to Libra-f, but a slight decline in clinical metrics. This can be attributed to the inclusion of VQA tasks, which encouraged the model to focus on more fine-grained and grounded information. While VQA emphasises the detailed description of the single symptom, the *Findings* section generation task requires a comprehensive overview of the CXR, encompassing multiple normal and abnormal findings. This shift in focus also impacted the $F1_{temp}$ score, as each finding typically involves its own temporal changes. Additionally, the reduction in the number of generated findings entities led to a decrease in the number of identified temporal entities. In the second stage, fine-tuned on the *Findings* dataset, Libra achieved the best results across all metrics, indicating that incorporating additional datasets in the first stage enhances Libra's understanding and reasoning of CXRs.

## 5 PERFORMANCE ANALYSIS

We used CXRs from the official test split to generate *Findings* sections and perform a qualitative analysis of Libra. Figure 2 illustrates a case where no true prior image was available for reference. For example, while the ground truth report only mentioned the presence of "sternal wires," Libra not only identified their presence but also provided details on the specific type. This demonstrates Libra can deliver more precise clinical information by not only recognizing conditions but also distinguishing the specific types of surgical interventions.

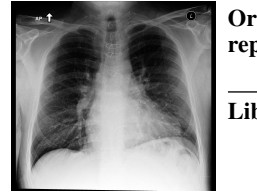

| **Original report** | The lungs are well expanded and clear. The cardiomediastinal silhouette, hilar contours, and pleural surfaces are normal. No pleural effusion or pneumothorax is present. Sternal wires are intact. |
|---|---|
| **Libra** | The lungs are well expanded and clear. The cardiomediastinal silhouette, hilar contours, and pleural surfaces are normal. No pleural effusion or pneumothorax is present. **Median sternotomy** wires are intact. |

Figure 2: Each identified radiological feature is highlighted. Surgical intervention type in **bold**.

For the samples with true prior images, Libra accurately describes the presence and changes in radiological conditions. As shown in the Figure 3, the current image revealed additional findings of pleural effusion and pneumonia compared to the prior image. The model effectively detected these progressive changes, provided detailed descriptions, and suggested potential causes for further investigation. Notably, it explicitly mentioned the comparative information relative to the prior image, offering a clear understanding of the temporal changes.

---

[5]For more ablation studies, including the impact of radiology-specific pre-trained models, see Appx. C.

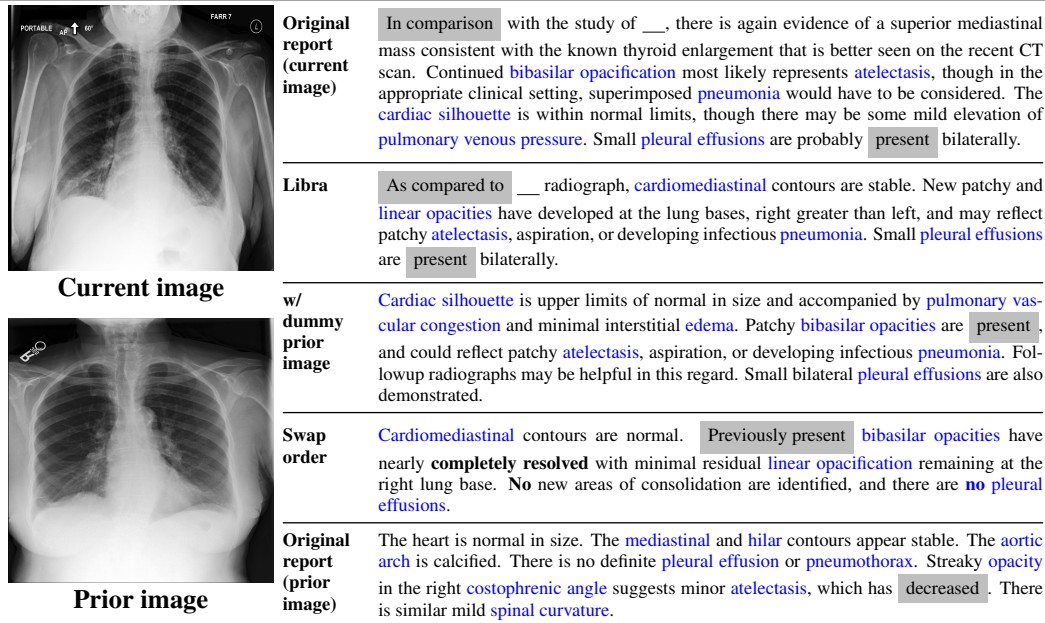

Figure 3: Radiological features are highlighted. Descriptions of temporal changes are marked. Key highlights from the swapped order are in **bold**. Heatmap analysis can be found in Appx. E.

To further assess the model's robustness, we conducted a more detailed analysis of this case. First, we generated a report using only the dummy prior image as a reference. While Libra successfully detected abnormal findings, it failed to specify changes relative to the previous image, describing only the current state without indicating disease progression. Moreover, it provided less clarity in distinguishing the distribution and severity of abnormalities. This suggests that, without a true prior image for comparison, the model struggles to accurately describe the progression of findings.

**Evaluating Temporal Consistency in Libra's Report Generation** To investigate the model's ability to capture temporal changes, we swapped the order of the images, using the true prior image as the current image and vice versa. Interestingly, the generated report indicated an improvement in the patient's condition, with most abnormalities resolved and no new findings detected, which is contrary to the ground truth of the original current image report. Moreover, the report generated from the swapped images closely resembled the original report of the true prior image, as shown at the bottom of Figure 3, which described a stable or improving condition for the patient. This striking observation demonstrates that Libra can effectively utilise both current and prior studies to generate accurate reports for assessing changes, simulating the conditions of standard clinical practice.

## 6 CONCLUSION

In this study, we presented Libra, a temporal-aware multimodal large language model specifically designed for chest X-ray report generation tasks. Trained exclusively on the open-access MIMIC-CXR dataset (Johnson et al., 2019b), Libra utilises a two-stage training framework that leverages a radiology-specific pre-trained image encoder and language model, connected through a specially designed Temporal Alignment Connector to bridge visual and textual modalities. By identifying observations from current CXR and referencing prior scan, Libra generates radiology reports with substantial improvements across all metrics compared to existing models of the same parameter scale. Qualitative analysis in the case study highlights Libra's ability to effectively utilise the temporal relationships between current and prior images, addressing the challenge of hallucinations related to prior study references in RRG tasks. Libra sets a new paradigm for MLLMs in multi-modal medical AI research. In the future, we aim to further enhance Libra's clinical applicability and accuracy, paving the way for the development of general-purpose AI models in the medical domain. For a detailed discussion of Libra's limitations and future work, please refer to Appx. A.3.

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

# A APPENDIX A

## A.1 RELATED WORK

**Radiology Report Generation** Several efforts have been made to develop models tailored for radiology report generation (RRG), driven by the need to address the long-tail distribution of observations in chest X-rays and provide fine-grained descriptions of findings. This task has gained prominence as an important objective for the designed system (Wang et al., 2018b).

Such systems inherently require a language generation component. Initially, recurrent neural networks were used (Liu et al., 2019), but these have recently been replaced by Transformer architectures (Miura et al., 2021b; Chen et al., 2022), including large language models (LLMs) such as PaLM (Chowdhery et al., 2022) and Vicuna-7B (Chiang et al., 2023).

To ensure clinical accuracy in generated text, some studies have moved beyond standard language modelling losses, instead employing reinforcement learning (RL) to optimise for rewards that prioritise "clinically relevant" features, such as the inclusion of specific findings (Liu et al., 2019; Irvin et al., 2019b) or maintaining logical consistency (Miura et al., 2021a; Delbrouck et al., 2022a). However, these methods often depend on external models like CheXbert (Smit et al., 2020) or RadGraph (Jain et al., 2021) to extract clinical entities, adding complexity to the optimisation process.

With the advancement of LLMs, an increasing number of studies have demonstrated that plain auto-regressive language modelling can achieve substantial results. However, we acknowledge that the benefits of more complex training objectives or RL-based approaches may be complementary. Consequently, research on leveraging temporal information in RRG tasks can be broadly categorised into two types: LLM-based methods and non-LLM-based methods.

**non-LLM-based Model** These architectures are typically transformer encoder-decoder models and their variants. They are often composed of multiple modules that require separate training. Meanwhile, they handle single and double image inputs by symbolically differentiating tasks and employing two distinct neural network architectures: one designed for single-image inputs and another for double-image inputs.

Serra et al. (2023) utilises symbolic alignment in its Longitudinal Projection Module and employs a separately trained BERT-based (Devlin et al., 2019) text generator. RECAP (Hou et al., 2023a) implements a transformer encoder-decoder with symbolic task differentiation and two-stage training, beginning with classification tasks followed by report generation. TiBiX (Sanjeev et al., 2024) adopts a transformer model with causal attention layers and incorporates learnable padding tokens to handle cases without prior images. BioViL-T (Bannur et al., 2023) is a self-supervised vision-language training framework that features a CNN–Transformer hybrid multi-image encoder trained jointly with a BERT-based text model.

On one hand, the difference in model parameter sizes, and on the other, as LLM-based models generally outperform other types of models in the RRG task, papers on non-LLM-based models or those using small language models typically do not compare their methods with LLM-based approaches. At the same time, we conducted comparisons and discussions to reaffirm this point, as detailed in Appx. D.2.

**LLM-based Model** Current LLM-based models have achieved notable success in the RRG task, largely benefiting from visual instruction tuning (Liu et al., 2023). Structurally, these models (Li et al., 2023a; Chaves et al., 2024; Hyland et al., 2024; Zhou et al., 2024; Park et al., 2024) typically consist of an image encoder and an adapter that bridges the connection to the LLM. These adapters often rely on single-layer hidden representations (e.g., the last or penultimate layer) from pre-trained image encoders, limiting their ability to simultaneously integrate information from multiple images.

In end-to-end training, LLM-based models handle multiple image inputs by concatenating them with textual prompts to form a composite input for the LLM. For example, the input format is "<Current Image Placeholder> + <Prior Imgae Placeholder> + <Prompt>".

Although our model adopts an end-to-end training paradigm, it does not rely on the vanilla approach to handle multiple images. Instead, we designed a novel adapter called the Temporal Alignment Connector (TAC). On one hand, TAC leverages all hidden layer features from the image encoder, setting it apart from the default configurations used in previous works. On the other hand, TAC models temporal information directly rather than simply concatenating two images in the latent space of the LLM.

**Radiological Image Representation** As radiology lies outside the distribution of most general-domain image models, a radiology-specific pre-trained model is crucial for the RRG task (Pérez-García et al., 2024).

Zhou et al. (2023) introduces a unified framework, Masked Record Modeling (MRM), which combines self-supervision and radiology report supervision to enhance radiograph representation learning. BioViL-T (Bannur et al., 2023) integrates a CNN-Transformer hybrid architecture for multimodal modelling, leveraging temporal structures in medical imaging and reports to perform tasks such as disease progression classification and report generation. BiomedCLIP (Zhang et al., 2024b) is a multimodal biomedical foundational model pre-trained on various biomedical tasks. RAD-DINO (Pérez-García et al., 2024) is a medical image encoder that employs a pure image-based self-supervised learning approach from DINOv2 (Oquab et al., 2024) for continuous pretraining, focusing exclusively on image data to avoid the limitations of text supervision.

In recent parallel works, RAD-DINO has been widely applied to RRG tasks, including MAIRA-2 (Bannur et al., 2024), M4CXR (Park et al., 2024). Furthermore, Pérez-García et al. (2024) highlighted that RAD-DINO outperforms other types of image encoders in RRG tasks. Therefore, our model also adopts RAD-DINO as the image encoder.

## A.2 RESEARCH OBJECTIVES

**Temporal Information** Temporal change is a crucial factor in understanding disease progression, especially in radiology, where paired images and corresponding radiology reports capture changes over time. This temporal context is often reflected in comparisons between current and prior scans, documenting symptom evolution or newly identified findings.

The relativity of position within the timeline is a key determinant of temporal information. This relative positioning does not alter the established description of well-established facts about symptoms already present in the current image. For instance, if the prior image is recent, descriptions of changes will be minimal, whereas an older prior image may highlight more pronounced changes.

**Research Object** Our study focuses on the chest X-ray (frontal view), rather than the individual patient. Specifically, we aim to generate a report for the current image while referencing the temporal information provided by a single prior image as auxiliary information.

Our model is primarily designed to handle scenarios with limited temporal information in the RRG task. For instance, in a case where a patient has two scans taken just milliseconds apart, the current and prior images would be nearly identical, as no pathological changes would manifest within such a short interval. This extreme scenario demonstrates how the model handles clinical practice under limited temporal information. In such cases, the correct diagnosis for this minimal interval would be that the patient's condition is "stable." In our design, the dummy prior image addresses scenarios where a true prior image is absent.

However, patients often undergo multiple prior scans, which may include temporal information from different orientations. This more complex temporal information lies beyond the scope of our current study. We will provide a detailed discussion of this scenario in Appx. A.3.

## A.3 LIMITATIONS AND FUTURE WORK

While our work represents a step forward in leveraging temporal information for radiology report generation, it also has several limitations that warrant further exploration.

**Handling Multiple Prior Scans** Our current model is designed to process a single prior scan alongside the current scan. While this approach aligns with standard clinical workflows, which typically prioritise the most recent prior study for comparisons, it overlooks scenarios where multiple prior scans could offer a richer temporal perspective. For instance, analysing a sequence of images spanning an extended period could provide deeper insights into gradual disease progression. Future efforts should focus on extending our framework to incorporate multiple prior scans efficiently, enabling a more nuanced understanding of temporal patterns in clinical data.

**Temporal Information Beyond Image Comparisons** At present, our model captures temporal information through paired image comparisons and their corresponding textual reports. However, clinical assessments often draw upon a broader context, including historical notes, laboratory results, and other longitudinal patient data. Expanding our approach to integrate these diverse temporal data sources could facilitate a more holistic understanding of disease trajectories and patient history, significantly enhancing clinical applicability.

**Challenges with Sparse Temporal Data** In cases where prior scans are unavailable or only minimally informative (e.g., scans taken within a very short time interval), our use of a "dummy prior image" addresses the lack of temporal data. However, the model's ability to interpret and generate meaningful outputs under these constraints may still be limited. Future work could explore methods for imputing or synthesising temporal context to address such cases more effectively.

**Computational Complexity** Our use of temporal alignment mechanisms and multi-layer feature integration introduces increased computational demands. This complexity poses challenges for real-time deployment, particularly in resource-constrained clinical environments.

**Generalisability Across Modalities and Datasets** This study focuses exclusively on frontal-view chest X-rays and utilises the MIMIC-CXR dataset (Johnson et al., 2019b). However, our approach has not been evaluated on other imaging modalities (e.g., CT or MRI) or datasets such as CheXpert (Irvin et al., 2019b) or PadChest (Bustos et al., 2020).

Based on the identified limitations, we propose several directions for future research and development:

- Develop frameworks for integrating multiple prior scans with dynamic temporal reasoning.

- Extend the model to handle multi-modal imaging and textual data for comprehensive diagnostics.

- Investigating the integration of temporal data from diverse clinical sources, including electronic health records (EHRs)

- Exploring lightweight model architectures for faster inference while maintaining high performance.

These enhancements will not only address the current limitations but also extend the potential of temporal-aware multimodal models in radiology and beyond.

# B APPENDIX B

## B.1 TRAINING CONFIGURATION

We train the model using a standard auto-regressive language modelling loss (cross-entropy). For this study, we utilise Meditron-7b (Chen et al., 2023b) as the LLM, with a total batch size of 16 throughout the training process. The training is conducted on a computational infrastructure equipped with A6000 GPU (48GB of memory) and configured with DeepSpeed ZeRO-3 optimization (Rajbhandari et al., 2020) and BF16 precision enabled. A cosine learning rate scheduler is employed, starting with a warm-up phase of $0.03$. In the first stage of training, we run for 1 epoch with a learning rate of $2 \times 10^{-5}$. In the second stage, the model is trained for 3 epochs at the same learning rate. The LoRA (Hu et al., 2021) parameters are set to $r = 128$ and $alpha = 256$. The final checkpoint for all runs is selected based on the observation of the minimum loss on the evaluation dataset throughout the training process.

### B.2 Detailed Description of Datasets

**MIMIC-CXR.** (Johnson et al., 2019b) This is a large, publicly accessible dataset comprising 377,110 DICOM images across 227,835 studies, each accompanied by a radiology report (Johnson et al., 2019b). For images, we use the commonly available JPEG files from MIMIC-CXR-JPG (Johnson et al., 2019a), rather than the original DICOM files, and we preprocess the dataset to exclude non-AP/PA scans. For each report, we extract the *Findings, Impression, Indication, History, Comparison,* and *Technique* sections using rule-based heuristics supported by the official MIMIC code repository (Johnson et al., 2018).

For the *Findings* section generation task, we discard studies where the *Findings* could not be extracted, while allowing for missing content in other clinical sections. The same approach is applied to the *Impression* section generation task. In all our experiments, we adhere to the official MIMIC-CXR dataset split. Additionally, we retrieve prior images by following the chronological order of studies as indicated by the official labels, selecting the closest prior study as the reference image. It is important to note that, to prevent data leakage between the train, validation, and test sets, prior images are retrieved only from within the same split.

**Medical-Diff-VQA.** (Hu et al., 2023) This dataset is a derivative of the MIMIC-CXR dataset, focused on identifying differences between pairs of main and reference images. The data split adheres to the original labelling, ensuring no data leakage occurs. In total, this dataset comprises 700,703 question-answer pairs derived from 164,324 main-reference image pairs. The questions are divided into seven categories: abnormality, location, type, view, presence, and difference. Each pair consists of a main (current) image and a reference (prior) image, both taken from different studies of the same patient. The reference image is always selected from an earlier visit, with the main image representing the later visit. Of the seven question types, the first six types focus on the main image, while the "difference" questions involve both images.

**MIMIC-Ext-*MIMIC-CXR-VQA*.** (Bae et al., 2023) This dataset is an extension of the MIMIC-CXR dataset, specifically designed for VQA tasks within the medical domain, specifically focused on CXRs. It includes questions generated from 48 unique templates covering seven content types: presence, anatomy, attribute, abnormality, size, plane, and gender. Each template was developed with the guidance of board-certified medical experts to ensure clinical relevance, addressing both standard medical VQA content and more complex logical scenarios. In total, the dataset consists of 377,391 unique entries. Since annotations are based on single images, the current image serves as a dummy prior image for all entries in our experiment.

For this study, we carefully selected datasets that provide complete reports and temporal information (i.e., prior images) to ensure alignment with our research objectives for the RRG task. After thoroughly evaluating other datasets, we found them unsuitable for the following reasons:

**CheXpert.** (Irvin et al., 2019b) This dataset includes annotated scans with label-specific annotations rather than full medical reports. While useful for training image encoders or annotation models, it is not appropriate for the RRG task, which requires complete diagnostic reports.

**PadChest.** (Bustos et al., 2020) Although it includes reports and corresponding prior images, its reports are in Spanish, placing cross-language training beyond the scope of our model.

**IU-Xray.** (Demner-Fushman et al., 2016) This dataset lacks patient-level metadata and prior study information, which is critical for our focus on temporal information in chest X-rays.

**Chest ImaGenome Dataset.** (Wu et al., 2021a)(a derivative of MIMIC-CXR) This dataset does not follow the official MIMIC-CXR split, risking data leakage across train, validation, and test sets.

### B.3 Evaluation Metrics Details

**Lexical Metrics** We employed common natural language generation metrics to quantify the overlap between generated and reference reports. Specifically, ROUGE-L (Lin, 2004) measures the length of the longest common subsequence between the generated and reference reports. BLEU-$\{1, 4\}$ (Papineni et al., 2002) calculates n-gram precision and applies a brevity penalty to discourage overly short predictions. METEOR (Banerjee & Lavie, 2005), computes the weighted harmonic mean of unigram precision and recall, with an additional penalty for fragmenting consecutive word

sequences. Finally, we report BERTScore (Zhang et al., 2020a), which leverages pre-trained contextual embeddings from BERT (Devlin et al., 2019) to match words in candidate and reference sentences based on cosine similarity. We used default parameters for all of these evaluation metrics.

For radiology-specific metrics, we used as many of the same evaluation scores as possible from previous studies (Tu et al., 2023; Hyland et al., 2024; Bannur et al., 2024; Chaves et al., 2024), including the following:

**RadGraph-based metrics** The RadGraph model (Jain et al., 2021) is designed to parse radiology reports into structured graphs. These graphs consist of clinical entities, which include references to anatomy and observations, as well as the relationships between these entities. This structured representation enables a more detailed and systematic analysis of radiology reports, facilitating downstream tasks such as information extraction, report generation, and clinical decision support.

These include RadGraph-F1 (Jain et al., 2021), which computes the overlap in entities and relations separately and then reports their average. And a variant of it, $RG_{ER}$ (Delbrouck et al., 2022b), which matches entities based on their text, type, and whether they have at least one relation[6].

**CheXpert F1** This set of metrics utilizes the CheXbert automatic labeler (Smit et al., 2020) to extract "present", "absent", or "uncertain" labels for each of the 14 CheXpert pathologies (Irvin et al., 2019a) from the generated reports and their corresponding references. In line with prior work, we report CheXpert-F1 for all 14 classes, as well as for the 5 most common findings in real-world CXR reports, referring to these as '[Macro/Micro]-F1-[5/14]'.

**CheXbert vector similarity** We also employ CheXbert vector similarity (Yu et al., 2022), which calculates the cosine similarity between the embeddings of the generated and reference reports after processing them through the CheXbert model (Smit et al., 2020).

**RadCliQ** In addition, we utilise RadCliQ (Radiology Report Clinical Quality), a composite metric that combines RadGraph-F1 and BLEU scores in a linear regression model to estimate the number of errors that radiologists are likely to detect in a report. To maintain consistency with previous research, we use version $0$ of it.

Both the CheXbert vector similarity, $RadCliQ_0$, and RadGraph-F1 metrics are calculated using the code released by Yu et al. (2022).

**Temporal Entity F1** The $F1_{temp}$, which is specifically designed to detect temporal entities. This score is neither a traditional lexical metric nor a adiology-specific metrics. Here is an example to help you understand better, as shown in Table 6.

Table 6: Evaluation of Candidate Using Temporal Entity F1

| Ground Truth | Candidate | ROUGE-L | RadGraph-F1 | $F1_{temp}$ |
|---|---|---|---|---|
| Compare with prior scan, pleural effusion has worsened. | The pleural effusion has progressively worsened since previous scan. | 0.47 | 0.86 | **1.0** |
| | The pleural effusion is noted again on the current scan. | 0.22 | 0.80 | **0.0** |

From these results, it is clear that the differences in lexical (ROUGE-L) and clinical (RadGraph-F1) metrics between the two candidates are relatively smaller compared to the $F1_{temp}$ score. This highlights that Temporal Entity F1 effectively captures and evaluates the quality of temporal information in radiology reports, distinguishing it more accurately than other standard metrics in the context of temporal information descriptions.

---

[6]$RG_{ER}$ is implemented as `F1RadGraph` with `reward=partial` by the radgraph package.

## B.4 PROMPT EXAMPLE

Here, we select examples from the MIMIC-CXR (Johnson et al., 2019b) dataset as instances and further synthesise them using GPT-4 (OpenAI et al., 2024) to avoid any ethical concerns, as shown in Table 7. Following Hyland et al. (2024)'s work, we adopt a rule-based method to extract relevant sections from the report of the current image, utilising a fixed system prompt and a dynamic clinical prompt for each current scan. We utilised four clinical instructions from the original report: {*Indication*}, {*History*}, {*Comparison*}, and {*Technique*}. Notably, unlike MAIRA-2 (Bannur et al., 2024), we do not utilise the report of the prior image.

Table 7: Examples of Prompt for Libra in RRG task.

| **Original Radiology Report** |
|---|
| EXAMINATION: Chest (Portable AP) |
| INDICATION: Dyspnea and cough, right-sided back pain. |
| HISTORY: Intubation with pulmonary edema. |
| COMPARISON: Chest radiographs on ___ and CT chest without contrast on ___. |
| TECHNIQUE: Portable upright chest radiograph. |
| FINDINGS: In comparison with the prior study, there are diffuse bilateral pulmonary opacifications, more prominent on the right. These findings could indicate severe pulmonary edema, but superimposed pneumonia or developing ARDS cannot be excluded. Monitoring and support devices are appropriately positioned. |

| **Prompt Content** |
|---|
| *[System prompt]* { |
| The assistant specialised in comparing Chest X-ray images, identifying differences, and noting temporal changes. |
| } |
| + |
| *<Image Representation Placeholder>* |
| + |
| *[Clinical prompt]* { |
| Provide a detailed description of the findings in the radiology image. Following clinical context: |
| Indication: Dyspnea and cough, right-sided back pain. |
| History: Intubation with pulmonary edema. |
| Comparison: Chest radiographs on ___ and CT chest without contrast on ___. |
| Technique: Portable upright chest radiograph. |
| } |

# C  APPENDIX C

## C.1  IMPACT OF THE TEMPORAL ALIGNMENT CONNECTOR UNDER GENERAL-DOMAIN PRE-TRAINED MODELS

Table 8: Results of ablation experiments for the Temporal Alignment Connector. '↓' indicates that lower is better. Values in (%) indicate the percentage decrease compared with the Libra-b.

| Metric | Libra-b | w/o TFM | w/o LFE | w/o PIPB | w/o TAC |
|---|---|---|---|---|---|
| **Lexical:** | | | | | |
| ROUGE-L | 27.26 | 26.80 (-1.69%) | 26.57 (-2.53%) | 27.00 (-0.95%) | 24.58 (-9.83%) |
| BLEU-1 | 34.94 | 33.61 (-3.81%) | 33.68 (-3.61%) | 34.47 (-1.35%) | 31.40 (-10.13%) |
| BLEU-4 | 11.74 | 10.97 (-6.56%) | 10.94 (-6.81%) | 11.57 (-1.45%) | 8.89 (-24.28%) |
| METEOR | 35.37 | 34.50 (-2.46%) | 34.30 (-3.03%) | 34.93 (-1.24%) | 32.41 (-8.37%) |
| BERTScore | 55.51 | 54.97 (-0.97%) | 54.75 (-1.37%) | 55.26 (-0.45%) | 53.07 (-4.40%) |
| $F1_{temp}$ | 24.77 | 23.54 (-4.97%) | 24.00 (-3.11%) | 24.68 (-0.36%) | 22.52 (-9.08%) |
| **Clinical:** | | | | | |
| RadGraph-F1 | 21.67 | 21.06 (-2.81%) | 20.73 (-4.34%) | 21.35 (-1.48%) | 19.77 (-8.77%) |
| $RG_{ER}$ | 26.28 | 25.45 (-3.16%) | 25.14 (-4.34%) | 25.84 (-1.67%) | 23.74 (-9.67%) |
| RadCliQ$_0$ (↓) | 3.17 | 3.20 (-0.95%) | 3.22 (-1.58%) | 3.18 (-0.32%) | 3.27 (-3.15%) |
| CheXbert vector | 39.58 | 38.74 (-2.12%) | 38.37 (-3.06%) | 39.49 (-0.23%) | 37.56 (-5.10%) |
| *CheXpert-F1:* | | | | | |
| Micro-F1-14 | 49.06 | 47.68 (-2.81%) | 47.57 (-3.04%) | 48.40 (-1.35%) | 46.57 (-5.08%) |
| Macro-F1-14 | 33.07 | 31.60 (-4.45%) | 31.78 (-3.90%) | 31.78 (-3.90%) | 31.27 (-5.44%) |
| Micro-F1-5 | 54.55 | 52.14 (-4.42%) | 52.94 (-2.95%) | 53.10 (-2.66%) | 50.72 (-7.02%) |
| Macro-F1-5 | 47.24 | 44.28 (-6.27%) | 44.39 (-6.04%) | 44.68 (-5.42%) | 43.48 (-7.96%) |

Domain-specific pre-trained models (i.e., RAD-DINO (Pérez-García et al., 2024) and Meditron (Chen et al., 2023b)) inherently incorporate domain-specific knowledge, such as phrasing conventions, pronoun usage, and even temporal information embedded in the training corpus.

To eliminate the influence of such factors, we used a general-domain image encoder (DINOv2 (Oquab et al., 2024)) and a LLM (Vicuna-7B-v1.5 (Chiang et al., 2023)), allowing the structural enhancements of TAC to be observed more directly. We adopted the same setup as the second ablation study in Section 4. We first conducted a baseline experiment, referred to as Libra-b, by fine-tuning only the adapter for the *Findings* generation task. As shown in Table 8, we then conducted ablation studies by sequentially removing different components from the model, including the TFM, LFE, Prior Image Prefix Bias (PIPB) in TFM, and the entire TAC. Removing TFM restricts the model to processing only the current image, using a configuration similar to LLaVA (Liu et al., 2023), but with a four-layer MLP to align the image representation with the LLM's hidden dimensions. It is important to note that without the TFM module, the model cannot process true prior images or dummy prior images, and is limited to only the current image as input. Without LFE, the model follows the LLaVA setup, using the penultimate layer of the image encoder to process single or paired images.

The ablation results are consistent with those observed when using domain-specific pre-trained models, as presented in Section 4. Compared to Libra-b, removing any TAC submodule results in a decline across all metrics. Removing the TFM caused a notable drop in the $F1_{temp}$ score (↓>4%), emphasising its role in capturing temporal information. The removal of the LFE significantly decreased RadGraph-related scores, highlighting its importance for extracting detailed image features. Removing the PIPB impacted clinical scores more than lexical scores. Finally, removing the TAC resulted in significant declines across all metrics, underscoring its critical role in integrating image details and temporal information.

## C.2  IMPACT OF THE TEMPORAL ALIGNMENT CONNECTOR AFTER THE SECOND STAGE

To further evaluate the impact of the Temporal Alignment Connector (TAC) on Libra's performance, we followed the setup of the second ablation study in Section 4. After the first stage of alignment, the model underwent a second stage of fine-tuning. This stage was designed to optimise the model's

Table 9: Results of ablation experiments for the Temporal Alignment Connector after the second stage. '↓' indicates that lower is better. Values in (%) indicate the percentage decrease compared with the Libra-2.

| Metric | Libra-2 | w/o TFM | w/o LFE | w/o PIPB | w/o TAC |
|---|---|---|---|---|---|
| **Lexical:** | | | | | |
| ROUGE-L | 35.31 | 35.16 (-0.42%) | 35.09 (-0.64%) | 35.23 (-0.23%) | 34.41 (-2.55%) |
| BLEU-1 | 49.92 | 49.44 (-0.97%) | 49.47 (-0.90%) | 49.75 (-0.34%) | 48.61 (-2.63%) |
| BLEU-4 | 23.05 | 22.67 (-1.66%) | 22.65 (-1.75%) | 22.97 (-0.35%) | 21.51 (-6.70%) |
| METEOR | 47.99 | 47.69 (-0.62%) | 47.62 (-0.77%) | 47.84 (-0.31%) | 46.95 (-2.16%) |
| BERTScore | 61.28 | 61.13 (-0.24%) | 61.07 (-0.34%) | 61.21 (-0.12%) | 60.60 (-1.12%) |
| $F1_{temp}$ | 33.52 | 33.10 (-1.27%) | 33.25 (-0.79%) | 33.49 (-0.09%) | 32.73 (-2.36%) |
| **Clinical:** | | | | | |
| RadGraph-F1 | 30.77 | 30.55 (-0.72%) | 30.43 (-1.10%) | 30.65 (-0.40%) | 30.07 (-2.27%) |
| $RG_{ER}$ | 35.44 | 35.16 (-0.79%) | 35.05 (-1.10%) | 35.29 (-0.42%) | 34.55 (-2.51%) |
| $RadCliQ_0$ (↓) | 2.83 | 2.84 (-0.35%) | 2.84 (-0.35%) | 2.85 (-0.71%) | 2.85 (-0.71%) |
| CheXbert vector | 45.32 | 45.08 (-0.53%) | 44.97 (-0.77%) | 45.27 (-0.11%) | 44.73 (-1.30%) |
| *CheXpert-F1:* | | | | | |
| Micro-F1-14 | 54.11 | 53.73 (-0.70%) | 53.70 (-0.76%) | 54.00 (-0.20%) | 53.41 (-1.30%) |
| Macro-F1-14 | 37.16 | 36.74 (-1.13%) | 36.78 (-1.02%) | 36.79 (-1.00%) | 36.64 (-1.40%) |
| Micro-F1-5 | 58.76 | 58.10 (-1.12%) | 58.32 (-0.75%) | 58.36 (-0.68%) | 57.65 (-1.89%) |
| Macro-F1-5 | 51.99 | 51.16 (-1.60%) | 51.19 (-1.54%) | 51.27 (-1.38%) | 50.87 (-2.15%) |

performance on the *Findings* section generation task by leveraging the aligned visual and textual features learned during the initial stage.

In this phase, we applied Low-Rank Adaptation (LoRA) (Hu et al., 2021) to fine-tune the pre-trained LLM (Meditron (Chen et al., 2023b)), while keeping the visual encoder (RAD-DINO (Pérez-García et al., 2024)) and TAC weights frozen. The baseline for this experiment is Libra-2 (in Table 9), which is derived from Libra-1 (in Table 4) after undergoing LoRA fine-tuning.

We conducted ablation studies by progressively removing different TAC components, including TFM, LFE, the Prior Image Prefix Bias (PIPB), and the entire TAC. The conclusions remain consistent with those in Section 4, demonstrating that removing any TAC submodule results in a decline across all metrics compared to Libra-2. This indicates that the performance improvements brought by TAC are stable and unaffected by changes in training stages. It further confirms that TAC has effectively embedded the capability to process temporal information within the model.

## C.3 ROBUSTNESS ASSESSMENT OF THE TEMPORAL ALIGNMENT CONNECTOR

Table 10: Results of ablation experiments for the Temporal Alignment Connector with additional LoRA fine-tuning after the second stage. '↓' indicates that lower is better. Values in (%) indicate the percentage decrease compared with the Libra-3.

| Metric | Libra-3 | w/o TFM | w/o LFE | w/o PIPB | w/o TAC |
|---|---|---|---|---|---|
| **Lexical:** | | | | | |
| ROUGE-L | 35.58 | 35.53 (-0.14%) | 35.51 (-0.21%) | 35.55 (-0.08%) | 35.28 (-0.86%) |
| BLEU-1 | 49.54 | 49.38 (-0.32%) | 49.39 (-0.30%) | 49.48 (-0.11%) | 49.10 (-0.88%) |
| BLEU-4 | 23.61 | 23.48 (-0.55%) | 23.47 (-0.58%) | 23.58 (-0.12%) | 23.07 (-2.28%) |
| METEOR | 47.61 | 47.51 (-0.21%) | 47.49 (-0.26%) | 47.56 (-0.10%) | 47.26 (-0.73%) |
| BERTScore | 61.54 | 61.49 (-0.08%) | 61.47 (-0.11%) | 61.52 (-0.04%) | 61.31 (-0.37%) |
| $F1_{temp}$ | 33.51 | 33.37 (-0.42%) | 33.42 (-0.27%) | 33.50 (-0.03%) | 33.24 (-0.79%) |
| **Clinical:** | | | | | |
| RadGraph-F1 | 29.82 | 29.75 (-0.24%) | 29.71 (-0.37%) | 29.78 (-0.13%) | 29.59 (-0.76%) |
| $RG_{ER}$ | 35.60 | 35.51 (-0.26%) | 35.47 (-0.37%) | 35.55 (-0.14%) | 35.30 (-0.84%) |
| $RadCliQ_0$ (↓) | 2.91 | 2.92 (-0.34%) | 2.92 (-0.34%) | 2.91 ( – ) | 2.93 (-0.68%) |
| CheXbert vector | 44.77 | 44.69 (-0.18%) | 44.65 (-0.26%) | 44.75 (-0.04%) | 44.57 (-0.45%) |
| *CheXpert-F1:* | | | | | |
| Micro-F1-14 | 52.45 | 52.33 (-0.23%) | 52.32 (-0.25%) | 52.41 (-0.08%) | 52.22 (-0.44%) |
| Macro-F1-14 | 30.77 | 30.65 (-0.38%) | 30.66 (-0.34%) | 30.67 (-0.33%) | 30.63 (-0.47%) |
| Micro-F1-5 | 54.42 | 54.22 (-0.38%) | 54.28 (-0.25%) | 54.30 (-0.23%) | 54.08 (-0.63%) |
| Macro-F1-5 | 44.58 | 44.34 (-0.54%) | 44.35 (-0.52%) | 44.37 (-0.46%) | 44.26 (-0.72%) |

In this experiment, we introduced an additional round of LoRA fine-tuning to induce overtraining and assess the robustness of the Temporal Alignment Connector (TAC). Following the setup in Appx. C.2, after integrating the first LoRA weights, we reinitialised a new set of LoRA adapters

for the LLM and conducted further training under the same second-stage fine-tuning configuration for one epoch. The baseline for this experiment is Libra-3 (as shown in Table 10), which is derived from Libra-2 (illustrated in Table 9) following this additional fine-tuning step.

The results reveal that, compared to Libra-2, Libra-3 demonstrates minimal changes in lexical scores, while clinical scores decline due to overfitting caused by the additional fine-tuning. Notably, the CheXpert (Smit et al., 2020) (Macro-F1-[5/14]) scores exhibit the most influential reduction.

However, the ablation studies reaffirm that the performance improvements introduced by TAC are robust and remain unaffected by variations in training strategies. This robustness is attributed to TAC's ability to effectively capture and preserve temporal image representations during the first training phase, which are subsequently preserved throughout further fine-tuning.

These findings highlight that TAC reliably embeds the capacity for temporal information processing into the model, making it a stable and essential component for handling temporal data in RRG tasks.

## C.4 IMPACT OF RADIOLOGY-SPECIFIC PRE-TRAINED MODELS

Table 11: Ablation results for radiology-specific pre-trained models in Libra. '↓' indicates that lower is better. Values in (%) indicate the percentage improvement compared to Libra-c.

| Metric | Libra-1 | w/o RadDINO | w/o Meditron | w/o RadDINO+Meditron |
|---|---|---|---|---|
| **Lexical:** | | | | |
| ROUGE-L | 27.56 | 27.66 (0.36%) | 27.29 (-0.98%) | 27.26 (-1.09%) |
| BLEU-1 | 34.84 | 35.32 (1.38%) | 34.91 (0.20%) | 34.94 (0.29%) |
| BLEU-4 | 11.51 | 12.56 (9.12%) | 11.61 (0.87%) | 11.74 (2.00%) |
| METEOR | 35.50 | 35.65 (0.42%) | 35.53 (0.08%) | 35.37 (-0.37%) |
| BERTScore | 55.87 | 55.89 (0.04%) | 55.58 (-0.52%) | 55.51 (-0.64%) |
| $F1_{temp}$ | 26.63 | 25.53 (-4.13%) | 24.78 (-6.95%) | 24.77 (-6.98%) |
| **Clinical:** | | | | |
| RadGraph-F1 | 22.52 | 22.11 (-1.82%) | 23.13 (2.71%) | 21.67 (-3.77%) |
| $RG_{ER}$ | 27.32 | 26.72 (-2.20%) | 27.53 (0.77%) | 26.28 (-3.81%) |
| $RadCliQ_0$ (↓) | 3.10 | 3.13 (-0.97%) | 3.08 (0.65%) | 3.17 (-2.26%) |
| CheXbert vector | 42.02 | 40.78 (-2.95%) | 41.94 (-0.19%) | 39.49 (-6.02%) |
| *CheXpert-F1:* | | | | |
| Micro-F1-14 | 52.84 | 51.55 (-2.44%) | 51.45 (-2.63%) | 49.06 (-7.15%) |
| Macro-F1-14 | 36.87 | 34.58 (-6.21%) | 37.20 (0.90%) | 33.07 (-10.31%) |
| Micro-F1-5 | 56.63 | 55.00 (-2.88%) | 55.39 (-2.19%) | 54.55 (-3.67%) |
| Macro-F1-5 | 49.33 | 47.26 (-4.20%) | 47.62 (-3.47%) | 47.24 (-4.24%) |

Aligning radiology images with textual information is a key challenge in RRG tasks. To demonstrate the benefits of using radiology-specific pre-trained models for more accurate feature representation and improved MLLM performance, we initialised a Libra model with RadDINO, the TAC, and Meditron-7b, conducting the first stage of training, denoted as Libra-1 (This is consistent with the baseline setup of the second ablation study in Section 4). Then we replaced the image encoder and LLM with their general-domain counterparts, DINOv2 and Vicuna-7B-v1.5, respectively. Finally, we replaced both components, which is also referred to as Libra-b.

As shown in Table 11, substituting radiology-specific pre-trained models with general-domain models resulted in a notable decline in clinical scores, while the impact on lexical scores was minimal. Notably, replacing the radiology-specific image encoder caused a more pronounced decline in clinical metrics compared to replacing the language model. This suggests that accurate medical image representation provides greater benefits in RRG tasks, indicating the importance of incorporating domain-specific knowledge into pre-trained models to enhance Libra's performance.

## C.5 INCREMENTAL STUDY

In this section, we conducted an incremental study of our model, mirroring the logical framework we used to construct Libra. We started with an architecture similar to LLaVA, comprising a pre-trained CLIP image encoder, a randomly initialised four-layer MLP as the adapter, and Vicuna-7B-v1.5 as the LLM. We train the adapter on the *Findings* section generation task to establish an initial baseline.

Gradual enhancements are then introduced, as shown in the Table 12. First, the image encoder is replaced with DINOv2. Building upon this, we integrate the LFE, the prefix module of the TAC connector. Subsequently, we add the TFM, the suffix module, thus completing the full TAC connector. Following this, the image encoder and LLM are progressively replaced with biomedical domain models, RAD-DINO and Meditron, respectively. The dataset used for pretraining in the first stage is then extended. Finally, the model is fine-tuned for downstream tasks to obtain Libra.

Table 12: Results of ablation experiments for key components of Libra on *Findings* section generation performance. $^*$ indicates our initialised model. $^/$ denotes component replacement. $^+$ signifies structural addition. $^\ddagger$ represents dataset expansion. The best performances are highlighted in **bold**, and the second-best scores are underlined. '↓' indicates that lower is better.

| Metric | Stage 1: Temporal Feature Alignment | | | | | | | Stage 2 |
|---|---|---|---|---|---|---|---|---|
| | $^*$**Initial** | $^/$**DINO** | $^+$**LFE** | $^+$**TFM** | $^/$**RAD-DINO** | $^/$**Meditron** | $^\ddagger$**Dataset** | **Libra** |
| **Lexical:** | | | | | | | | |
| ROUGE-L | 23.77 | 24.58 | 26.57 | 27.26 | 27.29 | 27.56 | 27.27 | **36.66** |
| BLEU-1 | 31.48 | 31.40 | 33.68 | 34.94 | 34.91 | 34.84 | 41.24 | **51.25** |
| BLEU-4 | 8.41 | 8.89 | 10.94 | 11.74 | 11.61 | 11.51 | 13.59 | **24.54** |
| METEOR | 32.1 | 32.41 | 34.3 | 35.37 | 35.53 | 35.50 | 39.44 | **48.90** |
| BERTScore | 52.76 | 53.07 | 54.75 | 55.51 | 55.58 | 55.87 | 56.00 | **62.50** |
| $F1_{temp}$ | 21.60 | 22.52 | 24.00 | 24.77 | 24.78 | 26.63 | 24.80 | **35.34** |
| **Clinical:** | | | | | | | | |
| RadGraph-F1 | 18.58 | 19.70 | 20.73 | 21.67 | 23.13 | 22.52 | 20.45 | **32.87** |
| $RG_{ER}$ | 23.05 | 23.74 | 25.14 | 26.28 | 27.53 | 27.32 | 25.19 | **37.57** |
| $RadCliQ_0$(↓) | 3.35 | 3.26 | 3.22 | 3.17 | 3.08 | 3.10 | 3.31 | **2.72** |
| CheXbert vector | 35.59 | 37.94 | 38.37 | 39.49 | 41.94 | 42.02 | 35.33 | **46.85** |
| *CheXpert-F1:* | | | | | | | | |
| Micro-F1-14 | 44.75 | 46.57 | 47.57 | 49.06 | 51.45 | 52.48 | 43.63 | **55.87** |
| Macro-F1-14 | 25.13 | 31.27 | 31.07 | 33.07 | 37.20 | 36.87 | 25.68 | **40.38** |
| Micro-F1-5 | 45.97 | 50.72 | 52.94 | 54.55 | 55.39 | 56.63 | 49.75 | **60.07** |
| Macro-F1-5 | 36.55 | 43.48 | 44.39 | 47.24 | 47.62 | 49.33 | 40.40 | **53.75** |

We observed that with each step of improvement, the overall performance consistently exceeded the previous state, demonstrating the essential role of each component in our optimisation scheme. Notably, in the alignment stage, the model showed the greatest overall improvement after the addition of the TFM, highlighting its capability to capture temporal information, which significantly enhanced performance on the RRA task.

However, after the data expansion optimisation in the first stage, the scores on lexical metrics improved, while clinical metrics exhibited a slight decline. This can be attributed to the introduction of the VQA task, which led the model to perceive more fine-grained grounded information. This is distinct from the report generation task, which focuses on describing the holistic level. Furthermore, this shift also affects the $F1_{temp}$ score, as temporal entities are often associated with the presence of specific symptoms. Ultimately, these dropped scores are improved again through the second stage of fine-tuning.

**Evaluation of Libra's Temporal Awareness.** Another approach to investigating the model's ability to capture temporal information is to evaluate it separately within the test split based on the presence or absence of true prior images, as shown in Table 13.

With the addition of the TFM, the model exhibited temporal awareness. It is worth noting that, for the first time, the $F1_{temp}$ score of samples with prior images surpassed those without, and this trend persisted through subsequent optimisations. This indicates that the structural enhancements have resulted in a sustained improvement in the model's temporal perception capabilities. An effective example is in Section 5.

Table 13: Results of ablation experiments for Libra on the $F1_{temp}$ score. Of the 2,461 official test samples, 2,117 include a prior image as a reference and 344 do not.

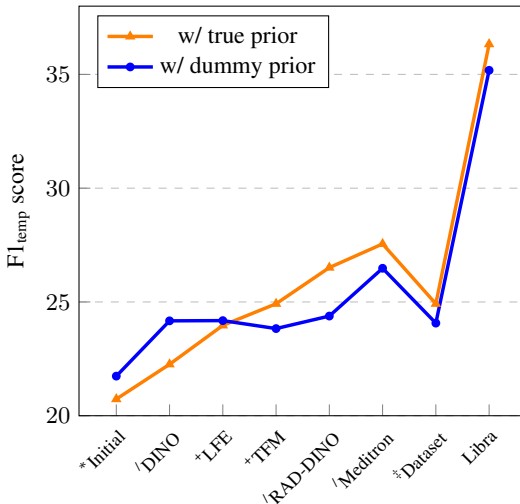

# D APPENDIX D

## D.1 DISCUSSION ON PERFORMANCE WITH RADIOLOGY FOUNDATION MODELS

Table 14: Findings Generation Performance of Libra and the Latest Concurrent Work. The best performances are highlighted in **bold**, and the second-best scores are underlined. '↓' indicates that lower values are better. The percentage (%) indicates the improvement over the best existing model.

| Metric | MedVersa | MAIRA-2 | M4CXR | Libra (%) |
|---|---|---|---|---|
| **Lexical:** | | | | |
| ROUGE-L | – | **38.4** | 28.5 | 36.7 (-4.4%) |
| BLEU-1 | – | 46.5 | 33.9 | **51.3** (10.3%) |
| BLEU-4 | 17.8 | 23.4 | 10.3 | **24.5** (4.7%) |
| METEOR | – | 42.0 | – | **48.9** (16.4%) |
| BERTScore | 49.7 | – | – | **62.5** (25.8%) |
| **Clinical:** | | | | |
| RadGraph-F1 | 28.0 | **34.6** | 21.8 | 32.9 (-4.9%) |
| $RG_{ER}$ | – | **39.7** | – | 37.6 (-5.3%) |
| $RadCliQ_0$(↓) | 2.7 | **2.6** | – | 2.7 (-3.8%) |
| CheXbert vector | 46.4 | **50.6** | – | 46.9 (-7.3%) |
| *CheXpert-F1:* | | | | |
| Micro-F1-14 | – | 58.5 | **60.6** | 55.9 (-7.8%) |
| Macro-F1-14 | – | **42.7** | 40.0 | 40.4 (-5.4%) |
| Micro-F1-5 | – | 58.9 | **61.8** | 60.1 (-2.8%) |
| Macro-F1-5 | – | 51.5 | 49.5 | **53.8** (4.5%) |

As shown in Table 14, these models belong to the category of radiology foundation models. Med-Versa (Zhou et al., 2024) and M4CXR (Park et al., 2024) support multiple tasks, including medical report generation, visual grounding, and visual question answering. In contrast, MAIRA-2 (Bannur et al., 2024) specialises in grounded radiology report generation, a novel task requiring image-level localisation for each identifiable finding or symptom. It is worth noting that the test sets differ slightly. All of these models utilise additional radiology information, such as lateral views, prior study reports of CXRs, or both.

Despite these considerations, Libra achieves the highest scores on most lexical metrics, including BLEU-{1, 4}, METEOR, and BERTScore, while trailing slightly behind MAIRA-2 on ROUGE-L. In clinical metrics, Libra predominantly ranks second, just behind the best-performing model. For clinical metrics, Libra consistently ranks second, just behind the top-performing model. In metrics that evaluate medical entities and their relationships, such as RadGraph-F1, $RG_{ER}$, and RadCliQ, Libra also ranks second. Similarly, Libra comes second in the CheXbert vector embedding score. However, in the CheXpert metrics, Libra ranks first in Macro-F1 for the 5-class subset, with only a slight dip in the Micro-F1 score for the 14-class subset.

Incorporating lateral images and prior study reports could enhance clinical scores. Additionally, strategies like chain-of-thought reasoning and grounded report generation further improve performance in RRG tasks. Looking ahead, we plan to develop model architectures that can automatically adapt to multiple tasks and diverse scenarios, enabling more efficient handling of additional radiological information.

## D.2 DISCUSSION ON PERFORMANCE WITH NON-LLM-BASED MODELS

Table 15: Findings Generation Performance of Libra and non-LLM-based Models. The best performances are highlighted in **bold**, and the second-best scores are underlined. These results are taken from the best performances reported in their original papers.

| Model | Lexical Metrics | | | | | | Clinical Metrics | | |
|---|---|---|---|---|---|---|---|---|---|
| | B-1 | B-2 | B-3 | B-4 | MTR | R-L | P | R | $F_1$ |
| R2Gen | 35.3 | 21.8 | 14.5 | 10.3 | 14.2 | 27.0 | 33.3 | 27.3 | 27.6 |
| R2GenCMN | 35.3 | 21.8 | 14.8 | 10.6 | 14.2 | 27.8 | 34.4 | 27.5 | 27.8 |
| $M^2$TR | 37.8 | 23.2 | 15.4 | 10.7 | 14.5 | 27.2 | 24.0 | 42.8 | 30.8 |
| KnowMAT | 36.3 | 22.8 | 15.6 | 11.5 | – | 28.4 | 45.8 | 34.8 | 37.1 |
| CMM-RL | 38.1 | 23.2 | 15.5 | 10.9 | 15.1 | 28.7 | 34.2 | 29.4 | 29.2 |
| CMCA | 36.0 | 22.7 | 15.6 | 11.7 | 14.8 | 28.7 | 44.4 | 29.7 | 35.6 |
| KiUT | 39.3 | 24.3 | 15.9 | 11.3 | 16.0 | 28.5 | 37.1 | 31.8 | 32.1 |
| DCL | – | – | – | 10.9 | 15.0 | 28.4 | 47.1 | 35.2 | 37.3 |
| METrans | 25.0 | 16.9 | 12.4 | 15.2 | – | 29.1 | 36.4 | 30.9 | 31.1 |
| ORGAN | 38.6 | 25.6 | 17.2 | 12.3 | 16.2 | 29.3 | 41.6 | 41.8 | 38.5 |
| COMG | 36.3 | 23.5 | 16.7 | 12.4 | 12.8 | 29.0 | – | – | – |
| MedM2G | 41.2 | 26.9 | 17.9 | 14.2 | – | 30.9 | – | – | – |
| BioViL-T | – | – | – | 9.2 | – | 29.6 | – | – | 17.5 |
| TiBiX | 32.4 | 23.4 | 18.5 | 15.7 | 16.2 | 33.1 | 30.0 | 22.4 | 25.0 |
| RECAP | 42.9 | 26.7 | 17.7 | 12.5 | 16.8 | 28.8 | 38.9 | 44.3 | 39.3 |
| Libra | **51.3** | **38.0** | **30.0** | **24.5** | **48.9** | **36.7** | **59.7** | **52.5** | **55.9** |

To facilitate comparison with non-LLM-based models, we selected evaluation metrics commonly used in these studies. These include BLEU-$\{1, 2, 3, 4\}$ (Papineni et al., 2002), METEOR (MTR) (Banerjee & Lavie, 2005), and ROUGE-L (R-L) (Lin, 2004). For clinical metrics, we adopted the CheXbert (Irvin et al., 2019a), reporting Precision (P), Recall (R), and $F_1$.

**Baseline** For performance evaluation, we compare our model with the following baselines: R2Gen (Chen et al., 2021), R2GenCMN (Chen et al., 2021), $M^2$TR (Nooralahzadeh et al., 2021), KnowMAT (Yang et al., 2022), CMM-RL (Qin & Song, 2022), CMCA (Song et al., 2022), KiUT (Huang et al., 2023), DCL (Li et al., 2023b), METrans (Wang et al., 2023b), ORGAN (Hou et al., 2023b), COMG (Gu et al., 2023), MedM2G (Zhan et al., 2024), BioViL-T (Bannur et al., 2023), TiBiX (Sanjeev et al., 2024), RECAP (Hou et al., 2023a).

As the results demonstrate, our model, like other LLM-based models, generally outperforms non-LLM-based models. This is attributed to advancements in LLMs and visual instruction tuning (Liu et al., 2023), enabling multimodal large language models (MLLMs) to excel in the RRG task.

# E   APPENDIX E

## E.1   TEMPORAL ALIGNMENT HEATMAP ANALYSIS AND FEATURE REPRESENTATION

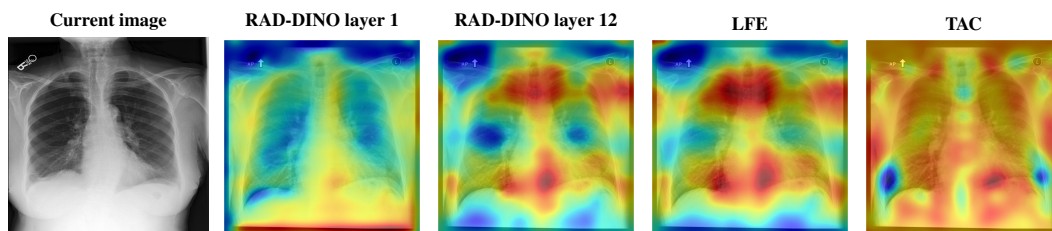

Figure 4: Heat map visualisation of image representations from different image encoder layers and the Temporal Alignment Connector (TAC), up-sampled using a Gaussian filter. Warm colours (red, yellow) indicate regions with higher weight allocations in the intermediate outputs of the "hidden-state" within the model blocks, while cool colours (blue, green) represent regions with lower weight.

The heatmap in Figure 4 corresponds to the example in Figure 2, where no true prior image was used as a reference. It highlights the clear differences in feature representations across layers of the RAD-DINO image encoder. The shallow layers primarily capture the overall lung structure, while the deeper layers focus on specific disease locations.

After being processed by the Layerwise Feature Extractor (LFE), the image feature representations show that larger symptom regions are assigned higher weights. This indicates that the image representation achieves a more fine-grained level of detail. Following the Temporal Alignment Connector (TAC), Libra integrates the weighted dummy prior image, resulting in a more uniform feature distribution. This reflects temporal information, indicating no significant changes compared to the prior study. This process helps the model provide a smoother image feature representation for subsequent processing by the text decoder (i.e., the LLM).

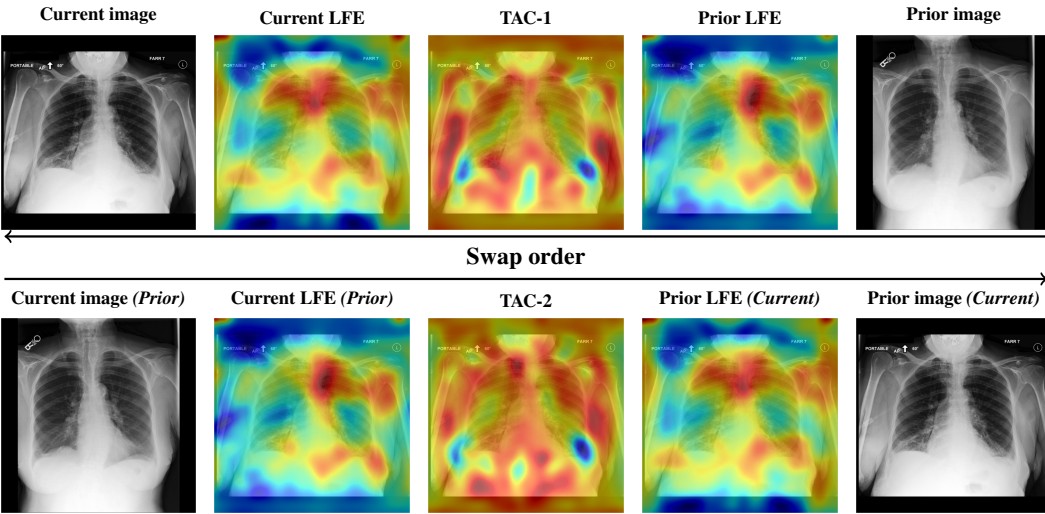

Figure 5: Heat map visualisation of image representations from the Temporal Alignment Connector (TAC), up-sampled using a Gaussian filter. The arrows ('→') represent the direction of temporal information, pointing from the true prior image to the true current image. Warm colours (red, yellow) indicate regions with higher weight allocations in the intermediate outputs of the "hidden-state" within the model blocks, while cool colours (blue, green) represent regions with lower weight.

The heatmap in Figure 5 corresponds to the example in Figure 3, where a true prior image is provided. As observed, after processing through the LFE, the model captures higher-granularity feature

representations in the symptom areas. After being processed by the TAC, the intermediate representation not only integrates these fine-grained features but also incorporates the difference information between the two images, reflecting the temporal information provided by the reference image, as shown in TAC-1 (Figure 5).

When the image order is swapped, treating the true prior image as the current image, the intermediate output from the LFE remains unchanged. Nevertheless, a comparison of the TAC-2 and TAC-1 outputs reveals noticeable differences in the lung feature representations. This demonstrates that the model's temporal perception is directional and that the TAC module successfully integrates temporal information from different time points. At the same time, it confirms that the LFE focuses solely on extracting image feature representations without encoding temporal aspects.

This result is expected because the residual connections in the TAC ensure that the current image remains the main modality, while the prior image serves as the auxiliary modality. When the order is swapped, the prior image becomes the main modality for subsequent processing. Thus, swapping the image order also changes the content of the generated report, as mentioned in Section 5. The primary difference lies in the reversal of the temporal state of symptoms, such as changing from "improving" to "worsening."

