# OpenReview forum: "Libra: Leveraging Temporal Images for Biomedical Radiology Analysis"
_ICLR.cc/2025/Conference — Submitted to ICLR 2025_

### Official Review · Reviewer_ryyo · 2024-10-18

**Soundness:** 3
**Presentation:** 2
**Contribution:** 2
**Rating:** 5
**Confidence:** 3

**Summary:**

This submission tackles the challenge of automated radiology report generation. Specifically, the authors aim to generate a radiology report’s Findings section given a current and (optionally) past radiology image of a patient. The paper proposes a method consisting of a pretrained image encoder, a new convolutional module designed to compress image features from the image encoder (termed “LFE”), an attention-based module designed to integrate information from two radiology images (termed “TFM”), and a pretrained language decoder. The main contributions are the two connecting modules, the “LFM” and “TFM”. Empirically, the method is shown to outperform baselines.

**Strengths:**

-	The need to integrate information from multiple images in generating a report is a real and well-motivated problem. The modules the authors propose are an interesting approach to doing so, and it seems this approach does improve the models’ ability to make past/present comparisons.
-	The authors propose a number of other changes in their radiology report pipeline, including an additional feature transformation module on top of the pretrained image encoder as well as a particular training scheme and prompt. The authors present a thorough ablation section to show each change improves performance. Empirically, the model is shown to outperform reasonable baselines.

**Weaknesses:**

-	The proposed approach is brittle regarding how many images can be input. With the current model, two scans are required to be input. Thus, when a patient does not have a prior scan, the authors propose using the current scan as a “dummy” prior scan instead. While many patients do not have previous scans, others often have multiple previous scans. Extending the proposed framework to account for multiple prior scans would increase computational complexity in the cross-attention head and I believe would require multiple “dummy” prior scans to be input for any patient who does not have the maximum number of scans. This inflexibility makes the proposed method less attractive for solving the posed problem of “temporally aware report generation” compared to methods that are more flexible with regard to number of input images.
-	The framing of the paper could be improved. The paper is pitched as a temporal alignment paper in the Abstract and Introduction, however it seems that most of the improvement in the paper come from a constellation of other changes made to the report generation pipeline. Specifically, the empirical evidence supporting their approach to using past+present images is not particularly strong. Most standard evaluation metrics only change 1-2% when using the past image. In contrast, it seems the additional complexity introduced by the LFE module (which integrates features from a pretrained image encoder), the additional layers in the TFM module, and other changes made to the training pipeline (the prompt, the use of VQA, etc.) impact performance more significantly, impacting performance metrics by 5-10%. Given the relative improvements of the authors’ contributions, I think the framing of the paper could be revised.
-	The paper’s writing could be improved to be more concise and clear; I found many sections overly wordy and others lacking detail.


-------------------------

I have read through the other reviewers' comments as well as the responses from the authors. While I appreciate the detailed responses the authors prepared, I do not find the clarifications particularly helpful or convincing and there is little new information presented in the rebuttal. As a result, I am leaving my review score unchanged and would vote to reject this submission.

**Questions:**

-	How long does inference take for a single study take, and on what hardware?
-	Is the choice to use features from throughout the image encoder ablated? This is proposed as one of two main contributions of the paper in the Introduction, however I do not see that choice evaluated anywhere. I see the ablations on the LFE, however you could ablate the use of multi-layer features from the encoder without removing the LFE.
-	A Limitations and Future Work section would be helpful additions. For example, the same approach could be used to integrate info from multiple imaging modalities, which could be an interesting avenue for future exploration.

---

> ### Author Response · Authors · 2024-11-20
> **Response to Reviewer ryyo (1/4)**
>
> Dear Reviewer ryyo,
>
> Thanks for your valuable feedback and constructive suggestions. We sincerely appreciate your recognition of our model's performance and comprehensive experimental analysis. Our detailed response to your concerns is listed as follows:
>
> ***
> **W1. The proposed approach is brittle regarding how many images can be input. With the current model, two scans are required to be input. Thus, when a patient does not have a prior scan, the authors propose using the current scan as a "dummy" prior scan instead. While many patients do not have previous scans, others often have multiple previous scans. Extending the proposed framework ... This inflexibility makes the proposed method less attractive for solving the posed problem of "temporally aware report generation" compared to methods that are more flexible with regard to number of input images.**
>
> **A1.**
>
> We greatly appreciate your insightful suggestions and extending regarding the challenges our model faces when processing multiple prior images. In the latest manuscript, we will highlight that our study on the RRG (Radiology Report Generation) task specifically **focuses** on chest X-rays to reduce ambiguity. Furthermore, we will include a detailed discussion of our model's limitations (e.g., handling multiple prior image scans) and future research directions.
>
> It is important to note that the research object in our study is the **chest X-rays** (frontal view), not the **patient**. While a patient may have multiple prior images, each X-ray has **only** one closest prior scan from a timeline perspective. As clarified in **Appendix A.2 (line 890)**, we select the closest prior image as the single reference based on the official timestamps.
>
> * In practice, clinicians rely on the latest scan in combination with prior images to inform their diagnoses. These prior images may include scans from multiple time points, and referencing scans taken at different intervals can result in varying descriptions of symptom changes. Such differences in comparisons are ultimately reflected in the diagnostic report.
>
> Our model is primarily designed to handle **temporal information from two scans taken at different time points**, which aligns with real-world clinical scenarios. The temporal information is derived from comparisons between scans taken at different time points. In our setup, the "dummy prior image" **(line 184)** effectively simulates the rare case where a patient undergoes multiple scans at the same time.
>
> * For example, in a specific case where a patient has two scans just milliseconds apart, the current and prior images would be nearly identical, as no pathological changes would manifest within such **a short interval**. This extreme scenario simulates how the model handles clinical practice under limited temporal information.
>
> In such case, the correct diagnosis for this tiny time interval would be that the patient's condition is **"stable"**. Thus, the dummy prior image addresses scenarios where a true prior image is absent. This approach improves the model's robustness by training it to correctly interpret temporal information even in cases with very short time intervals.
>
> As shown in **Section 2.1 (Figure 1)**, our model architecture always processes two images and learns the temporal information embedded in the data set during end-to-end training. The residual connections in the TAC structure ensure that the current image always serves as the primary modality, while the prior image or dummy prior image acts as an auxiliary modality.
>
> * Based on our research object, the **solution for patients with multiple scans** is straightforward: the model only needs the most current scan as the primary input and the oldest prior scan as a reference. This approach can generate reports that relatively accurately reflect the patient's current condition.
>
> * The **position relativity** in the timeline is the key factor affecting temporal information, and this relative positioning does not alter the description of **well-established facts** about symptoms already present in the current image.
>
> The generated report not only diagnoses symptoms but also describes **trends** in disease progression. For instance, if the prior image is recent, the description of changes would be minimal, whereas a much older prior image would result in a report with more significant changes. This approach aligns closely with how clinicians assess and diagnose patients in practice.
>
> Patients undergoing multiple prior scans often include temporal information in different orientations, which is currently beyond the scope of our study. In the revised manuscript, we will add this thought-provoking suggestion to the **Limitations and Future Work** section, further discussing the model's performance in different clinical scenarios. We welcome any specific suggestions regarding the challenges faced by the model.

---

> ### Author Response · Authors · 2024-11-20
> **Response to Reviewer ryyo (2/4)**
>
> ***
> **W2. The framing of the paper could be improved. The paper is pitched as a temporal alignment paper in the Abstract and Introduction, however it seems that most of the improvement in the paper come from a constellation of other changes made to the report generation pipeline. Specifically, the empirical evidence supporting their approach to using past+present images is not particularly strong. Most standard evaluation metrics only change 1-2% when using the past image. In contrast, it seems the additional complexity introduced by the LFE module (which integrates features from a pretrained image encoder), the additional layers in the TFM module, and other changes made to the training pipeline (the prompt, the use of VQA, etc.) impact performance more significantly, impacting performance metrics by 5-10%. Given the relative improvements of the authors’ contributions, I think the framing of the paper could be revised.**
>
> > **Review:** The framing of the paper could be improved. The paper is pitched as a temporal alignment paper in the Abstract and Introduction, however it seems that most of the improvement in the paper come from a constellation of other changes made to the report generation pipeline. Specifically, the empirical evidence supporting their approach to using past+present images is not particularly strong.
>
> **A2 (Answer-1).**
>
> We sincerely appreciate your suggestions for enhancing the readability and clarity of our manuscript, which will make it stronger! In the revised manuscript, we will emphasise that our model **focuses** on temporal alignment by aligning images and text while capturing temporal information between images, rather than improving the entire report generation pipeline, to reduce ambiguity for readers.
>
> Our model differs from existing methods in terms of vision-language alignment for temporal information on RRG. As noted in **line 60**, current parallel works employ the adapter that only accept a single image as input. Meanwhile, these adapters rely on a single hidden layer representation from the pre-trained encoder (either the last or penultimate layer) and cannot integrate information from multiple images simultaneously. The only way these MLLMs handle multiple image inputs is by concatenating multiple images and text as a combined input to the LLM, for example, *"<image 1 placeholder>+<image 2 placeholder>+<prompt>"*.
>
> * In contrast, our model, as described in **Section 2.2.1**, leverages all hidden layer features from the image encoder, which distinguishes it from previous works. This approach allows our model to use these multi-layered features for higher granularity in feature representation.
> * Additionally, our Temporal Alignment Connector (TAC) consistently receives **two** images as input. As shown in **Section 2.2.2**, *$Equation (3-6)$*, the residual connections allow the prior image or dummy prior image to serve as an auxiliary modality, complementing the current image. In end-to-end training, the model aligns not only image and text representations but also incorporates temporal information, setting it apart from existing methods.
>
> Meanwhile, to ensure a rigorous statement, we will replace **“visual language models (VLMs)”** with **“multimodal large language models (MLLMs)”** in the revised manuscript. We acknowledge that VLMs broadly cover multimodal models that learn from images and text, including large language models (LLMs) based models, transformer encoder-decoder models, and their variants. As LLM-based models typically outperform other types of models on RRG tasks, we primarily compare Libra with other LLM-based models to ensure a fair evaluation.
>
> In the latest manuscript, we will include a detailed discussion of prior work to more clearly highlight the innovations our model brings to the RRG task.

---

> ### Author Response · Authors · 2024-11-20
> **Response to Reviewer ryyo (3/4)**
>
> (Follow-Up to **A2**)
>
> > **Review:** Most standard evaluation metrics only change 1-2% when using the past image. In contrast, it seems the additional complexity introduced by the LFE module (which integrates features from a pretrained image encoder), the additional layers in the TFM module, and other changes made to the training pipeline (the prompt, the use of VQA, etc.) impact performance more significantly, impacting performance metrics by 5-10%. Given the relative improvements of the authors’ contributions, I think the framing of the paper could be revised.
>
> **A2 (Answer-2).**
>
> As you described, "evaluation metrics only change 1-2% when using the past image", such results are **expected** (as shown in **Section 4, Table 3**).
>
> In **Section 4**, the first ablation study is **not** designed to evaluate the TAC's ability to perceive temporal information. Instead, this experiment focuses on model inference. In the control group, all samples use a dummy prior image. Our model does **not** use a symbolic way to distinguish between the presence or absence of a prior image; it always receives two images as input.
>
> * Through end-to-end training, the model has already acquired the ability to perceive temporal information. Therefore, whether using a true prior image or a dummy prior image during inference, the generated report will contain the relevant temporal information.
>
> * In the control experiment (in **Section 4, Table 3**) where only a dummy prior is used, all test samples are treated as **"stable"**. In this setup, our model still generates a report based on the current image while referencing the prior (dummy) image. The generated report thus describes pathological features **without** reflecting true temporal changes about corresponding symptoms.
>
> As shown in **Table 3**, when the model uses prior images, only the $F1_{temp}$  score shows a substantial improvement **(8%)**, while other evaluation metrics change by only **1-2%**. This result also aligns with the expectations of this ablation study.
>
> * This is because the $F1_{temp}$ score is **better** suited for evaluating these descriptions of change, whereas other metrics focus on overall report quality and domain-specific assessments. Those metrics **struggle** to quantify the impact of temporal information on overall quality.
>
> In **Section 3.2**, we provide a detailed explanation of the $F1_{temp}$  metric, which is specifically designed to detect temporal entities. This score is neither a traditional lexical metric nor a clinical metric.  **Appendix A.3** provides a detailed description of the focus of these other evaluation metrics. Here is an example to help you better understand $F1_{temp}$.
>
> * Gound truth: "Compare with prior scan, pleural effusion has worsened."
> * Candidate 1: "The pleural effusion has progressively worsened since previous scan."
>   * Candidate 1 scores: ROUGE-L=0.47 RadGraph-F1=0.86  $F1_{temp}$=**1.0**
> * Candidate 2: "The pleural effusion is noted again on the current scan."
>   * Candidate 2 scores: ROUGE-L=0.22  RadGraph-F1=0.80  $F1_{temp}$=**0.0**
>
> From this results, it is evident that the differences in lexical (ROUGE-L) and clinical (RadGraph-F1) metrics between the two candidates are relatively smaller compared to the $F1_{temp}$  score. This highlights that the $F1_{temp}$ metric effectively distinguishes the temporal information quality between the two references. This explains why, in **Section 4 (Table 3)**, only the $F1_{temp}$ score shows a substantial improvement.
>
> In the revised manuscript, we will include a detailed explanation of the $F1_{temp}$ metric and the specific example mentioned above. Additionally, we will swap the first and second ablation studies (in **Section 4**) to revise the framing of the paper, aligning it better with the relative improvements of our contributions.
>
> ***
> **W3. The paper's writing could be improved to be more concise and clear; I found many sections overly wordy and others lacking detail.**
>
> **A3.**
>
> We sincerely appreciate your feedback regarding the writing style. We acknowledge that some sections appear overly wordy, while others might lack sufficient detail. We will carefully revise the manuscript to ensure greater clarity and conciseness, while balancing details where necessary (as mentioned in the **previous two answers**). These improvements will be reflected in the revised manuscript.

---

> ### Author Response · Authors · 2024-11-20
> **Response to Reviewer ryyo (4/4)**
>
> ***
> **Q1. How long does inference take for a single study take, and on what hardware?**
>
> **A-Q1.**
>
> Inference for a single study and generating a report takes approximately 1-2 seconds (in 4-bit quantization, which includes both inference and text decoding). The specific hardware details are provided in **Appendix A.1**. We used a single A6000 GPU (48GB of memory) for both training and inference.
>
> ***
> **Q2. Is the choice to use features from throughout the image encoder ablated? This is proposed as one of two main contributions of the paper in the Introduction, however I do not see that choice evaluated anywhere. I see the ablations on the LFE, however you could ablate the use of multi-layer features from the encoder without removing the LFE.**
>
> **A-Q2.**
>
> **Without the LFE module**, it is equivalent to conducting an ablation study on the multi-layer feature representations throughout the image encoder.
>
> In **Section 2.2.1**, *$Equation (1)$* provides a detailed description of the LFE module, which compresses multi-layer image representations.
> * The size-compression pattern of the convolutional layers is **directly related to** the number of hidden layers in the image encoder. The "use of multi-layer features" is a functionality enabled by the **LFE** module.
> * Without the LFE module, the model **lacks** the structure to compress multi-layer representations from the image encoder, ultimately failing to align them with the hidden dimensions of the LLM.
> * In Section 2.2.2, Equations (3-7) show that the **TFM** module is solely responsible for temporal alignment and **cannot** handle multi-layer feature embeddings.
>
> In **Section 4 (line 434)**, we mention that the final configuration mirrors LLaVA [a] but uses a four-layer MLP. In LLaVA, the setup utilises the **penultimate** layer from the image encoder. We acknowledge the ambiguity here and will add a clarification in the latest manuscript:
> > *"It is important to note that without the LFE, the model also follows the LLaVA setup, using the penultimate layer of image encoder."*
>
> Furthermore, the ablation study on the different single-layer representations of the image encoder in the RRG task is currently beyond the scope of our study.
>
> *[a] Liu, Haotian, et al. "Improved baselines with visual instruction tuning." Proceedings of the IEEE/CVF Conference on Computer Vision and Pattern Recognition. 2024.*
>
> ***
> **Q3. A Limitations and Future Work section would be helpful additions. For example, the same approach could be used to integrate info from multiple imaging modalities, which could be an interesting avenue for future exploration.**
>
> **A-Q3.**
>
> We sincerely appreciate your constructive suggestions. We acknowledge that the **Limitations and Future Work** section is highly valuable. In the revised manuscript, we will discuss potential model extensions, as mentioned in **"A-1"**, such as handling multiple prior images from different time points and incorporating previous reports.
>
> ***
> We sincerely hope that our responses have addressed your concerns and hope you will consider increasing your score. If we have left any notable points of concern overlooked, we would greatly appreciate your feedback, and we will attend to these points. Additionally, we will incorporate all the suggestions and discussions mentioned above in the latest manuscript. Thanks again for your thoughtful review and consideration.

---

> ### Author Response · Authors · 2024-11-25
>
> Dear Reviewer ryyo,
>
>
> Thank you very much for taking the time to review our paper.
>
> We have made every response to address the issues raised by each reviewer and have provided corresponding sections in the latest manuscript. As no new ablation studies or comparisons were requested, no new information has been included.
>
>
> Best regards,
>
> Authors

---

### Official Review · Reviewer_y9E4 · 2024-10-30

**Soundness:** 2
**Presentation:** 2
**Contribution:** 2
**Rating:** 6
**Confidence:** 4

**Summary:**

This paper introduces Libra, a novel Visual-Language Model (VLM) designed for radiology report generation (RRG) that leverages temporal imaging data to improve report accuracy. To address disease progression in radiology, Libra integrates a radiology-specific image encoder with a medical language model and introduces a Temporal Alignment Connector (TAC). This connector comprises a Layerwise Feature Extractor (LFE) to capture high-granularity imaging features and a Temporal Fusion Module (TFM) for synthesizing temporal references.

**Strengths:**

1. The paper presents an innovative approach for handling prior study citations across various time points in report generation tasks.

2. The development of the Temporal Alignment Connector showcases a sophisticated method for capturing and integrating temporal information across multiple images.

3. A comprehensive experimental analysis, including ablation studies and qualitative comparisons, is provided to validate the effectiveness of the proposed methods.

**Weaknesses:**

1. Comparative Results: The comparative results do not convincingly demonstrate Libra's superiority. Although MIMIC-Diff-VQA is derived from MIMIC-CXR, the comparison seems unbalanced, as Libra was trained on both MIMIC-CXR and MIMIC-Diff-VQA, while the other model was only trained on MIMIC-CXR.

2. Effectiveness of the Temporal Alignment Connector: The authors overstate the effectiveness of the Temporal Alignment Connector (e.g., "significant enhancements across all metrics" in line 398). While the Temporal Fusion Module aims to provide temporal-aware representations, the ablation results in Table 3 (using dummy images) yield comparable results, with the exception of the F1 score. If Libra effectively perceives and utilizes temporal information, it would be beneficial to explain why incorporating true prior images results in only marginal improvement.

3. Experimental Settings: Some experimental settings deviate from standard practices. For instance, in Q2 of Section 4, the encoder and LLM should align with those used in Libra (i.e., RAD-DINO and Meditron).

4. Clarity in Table 5: Table 5 lacks clarity. Notably, using only Findings in Stage 1 yields better performance in Clinical Efficacy. One would intuitively expect the VQA dataset, which emphasizes detailed symptom descriptions, to enhance Clinical Efficacy, yet the results in Table 3 show a decrease.

5. Explanation of Figure 5: Provide more detailed explanations for Figure 5 in the Appendix, where the heatmap indicates that the Prior LFE shows more focused attention.

**Questions:**

See the Weaknesses part.

---

> ### Author Response · Authors · 2024-11-20
> **Response to Reviewer y9E4 (1/4)**
>
> Dear Reviewer y9E4,
>
> Thanks for your valuable feedback and constructive suggestions. We sincerely appreciate your acknowledgement of our innovation and comprehensive experimental analysis. Our detailed response to your concerns is listed as follows:
>
> ***
> **W1. Comparative Results: The comparative results do not convincingly demonstrate Libra's superiority. Although MIMIC-Diff-VQA is derived from MIMIC-CXR, the comparison seems unbalanced, as Libra was trained on both MIMIC-CXR and MIMIC-Diff-VQA, while the other model was only trained on MIMIC-CXR.**
>
> **A1.**
>
> We sincerely appreciate your suggestions for improving the comparability of the results in our manuscript, which will make it stronger! Our results allow for fair comparisons with other models.
>
> Since **all** datasets used in our study, including their variants, are derived from **MIMIC-CXR** [e] and strictly adhere to the official splits, no additional text augmentation was applied to the train/valid/test sets. Furthermore, as noted in **Appendix A.2 (line 891)**, prior image selection also follows the official labels, selecting only from within each respective split, which ensures that temporal information is not leaked. This setup ensures that our approach enables fair comparisons with other models.
>
> As mentioned in **Section 3.1 (line 267)**, for multimodal large language models (MLLMs), the first phase of alignment requires substantially larger and broader datasets compared to the fine-tuning stages in the second or third phases. Among the models we compared, several followed similar practices.
>
> * LLaVA-Rad [a], in the first phase, used additional data from CXR-697K, which includes sources like CheXpert, BraX, MIMIC-CXR, and PadChest (notably, PadChest is a Spanish-language dataset).
> * Med-PaLM [b], as a foundation model, utilised even more external data, including radiology-focused datasets such as VQA-RAD and Slake-VQA.
> * MAIRA-1 [c] adopted a different approach by using GPT to paraphrase reports in the MIMIC-CXR training split while keeping the validation and test sets unchanged.
> * MAIRA-2 [d], a model designed for grounded radiology report generation, also used additional datasets, including PadChest, USMix, and IU-Xray.
>
> For our model, in the first training stage, we used MIMIC-CXR and its variant, **MIMIC-Diff-VQA** [f]. This variant dataset breaks each report into various types of vision questions, such as "Is this PA view? - yes," providing finer-grained question-answer pairs for each image. As mentioned in **Appendix A.2 (line 895)**, MIMIC-Diff-VQA strictly adheres to the official MIMIC-CXR split labels, so there is **no** risk of data leakage between datasets. In the second training stage, we fine-tuned our model on MIMIC-CXR solely for the RRG task.
>
> Meanwhile, to ensure a rigorous statement, we will replace **“visual language models (VLMs)”** with **“multimodal large language models (MLLMs)”** in the revised manuscript. We acknowledge that VLMs broadly cover multimodal models that learn from images and text, including large language models (LLMs) based models, transformer encoder-decoder models, and their variants. As LLM-based models typically outperform other types of models on RRG tasks, we primarily compare Libra with other LLM-based models. We also acknowledge that our explanation of prior work on RRG is insufficient.
>
> * LLM-based models are typically trained in an end-to-end manner. In contrast, non-LLM-based models use a different architecture, generally comprising multiple modules that require separate training.
> * Meanwhile, non-LLM-based models handle single and double image inputs by symbolically differentiating tasks and employing two separate neural network architectures: one for single-image inputs and another for double-image inputs.
>
> Given these differences, **non-LLM-based** models are typically not compared with those of **LLM-based** models in their paper. In the revised manuscript, we will include a more detailed discussion of prior work, covering both the **models** and the **datasets** used for their training. This will provide readers with a comprehensive understanding of the latest advancements in the RRG (Radiology Report Generation) task and enhance the fairness and credibility of comparisons.
>
>
> *[a] Chaves, Juan Manuel Zambrano, et al. "Towards a clinically accessible radiology foundation model: open-access and lightweight, with automated evaluation."*
>
> *[b] Tu, Tao, et al. "Towards generalist biomedical AI."*
>
> *[c] Hyland, Stephanie L., et al. "Maira-1: A specialised large multimodal model for radiology report generation."*
>
> *[d] Bannur, Shruthi, et al. "Maira-2: Grounded radiology report generation."*
>
> *[e] Johnson, Alistair EW, et al. "MIMIC-CXR, a de-identified publicly available database of chest radiographs with free-text reports."*
>
> *[f] Hu, Xinyue, et al. "Expert knowledge-aware image difference graph representation learning for difference-aware medical visual question answering."*

---

> ### Author Response · Authors · 2024-11-20
> **Response to Reviewer y9E4 (2/4)**
>
> ***
> **W2. Effectiveness of the Temporal Alignment Connector: The authors overstate the effectiveness of the Temporal Alignment Connector (e.g., "significant enhancements across all metrics" in line 398). While the Temporal Fusion Module aims to provide temporal-aware representations, the ablation results in Table 3 (using dummy images) yield comparable results, with the exception of the F1 score. If Libra effectively perceives and utilizes temporal information, it would be beneficial to explain why incorporating true prior images results in only marginal improvement.**
>
> **A2.**
>
> We acknowledge that the use of the term "significant" here lacks statistical accuracy, and we will revise it in the latest manuscript. As you mentioned, "$F1_{temp}$ metric are comparable, while other metrics show only marginal improvement." Such results are as **expected**.
>
> In **Section 3.2**, we provide a detailed explanation of the $F1_{temp}$ metric, which is specifically designed to detect temporal entities. This score is neither a traditional lexical metric nor a clinical metric, as other metrics are not suited to capturing temporal information. **Appendix A.3** provides a detailed description of the focus of these other evaluation metrics. Here is an example to help you better understand $F1_{temp}$.
>
> * Gound truth: "Compare with prior scan, pleural effusion has worsened."
> * Candidate 1: "The pleural effusion has progressively worsened since previous scan."
>   * Candidate 1 scores: ROUGE-L=0.47 RadGraph-F1=0.86  $F1_{temp}$=**1.0**
> * Candidate 2: "The pleural effusion is noted again on the current scan."
>   * Candidate 2 scores: ROUGE-L=0.22  RadGraph-F1=0.80  $F1_{temp}$=**0.0**
>
> From this example, it is evident that the differences in lexical (ROUGE-L) and clinical (RadGraph-F1) metrics between the two candidates are relatively **smaller** compared to the $F1_{temp}$  score. This highlights that the $F1_{temp}$ metric effectively distinguishes the temporal information quality between the two references. This explains why, in **Section 4 (Table 3)**, only the $F1_{temp}$ score shows a substantial improvement.
>
> In **Section 4**, the first ablation study is **not** designed to evaluate the Temporal Alignment Connector's ability to perceive temporal information. Instead, this experiment focuses on model inference. In the control group, all samples use a dummy prior image. Our model does **not** use a symbolic way to distinguish between the presence or absence of a prior image; it always receives two images as input.
>
> * Through end-to-end training, our model has already acquired the ability to perceive temporal information. Therefore, whether using a true prior image or a dummy prior image during inference, the generated report will contain the relevant temporal information.
> * The dummy prior image can be seen as an extreme example, where a patient has two scans just milliseconds apart, the current and prior images would be nearly identical, as no pathological changes would manifest within such **a short interval**. In such case, the correct diagnosis for this tiny time interval would be that the patient's condition is **"stable"**. Thus, the dummy prior image also addresses scenarios where a true prior image is absent.
>
> In the control experiment (**in Table 3**) where only a dummy prior is used, all test samples are treated as **"stable"**. In this setup, our model still generates a report based on the current image while referencing the prior (dummy) image. The generated report thus describes pathological features **without** reflecting true temporal changes about corresponding symptoms.
>
> The $F1_{temp}$ score is **better** suited for evaluating these descriptions of change, whereas other metrics focus on overall report quality and domain-specific assessments. Those metrics **struggle** to quantify the impact of temporal information on overall quality, which explains why they show **only marginal improvement**. We will include a detailed explanation of the $F1_{temp}$ metric and the specific example mentioned above in the revised manuscript.

---

> ### Author Response · Authors · 2024-11-20
> **Response to Reviewer y9E4 (3/4)**
>
> ***
> **W3. Experimental Settings: Some experimental settings deviate from standard practices. For instance, in Q2 of Section 4, the encoder and LLM should align with those used in Libra (i.e., RAD-DINO and Meditron).**
>
> **A3.**
>
> We sincerely appreciate your suggestions regarding the ablation study in **Section 4(Q2)**, which will make it stronger! In the revised manuscript, we will provide results using encoders and LLMs consistent with those in Libra (i.e., RAD-DINO [g] and Meditron [h]). Additionally, the original **Q2** will be moved to the Appendix as an additional ablation study.
>
> The original ablation study in **Section 4(Q2)** focused on evaluating the performance improvements brought by the Temporal Alignment Connector (TAC). To eliminate the influence of **other factors**, we selected a general-domain image encoder and a LLM, ensuring that the structural enhancements of TAC could be observed more directly.
>
> * Domain-specific pre-trained models (i.e., RAD-DINO and Meditron) inherently incorporate domain-specific knowledge, such as phrasing habits, pronoun usage, and even embedded temporal information from the corpus. Considering these factors, the original ablation study focused on the TAC structure.
> * In **Appendix B.1**, we conducted an ablation study using radiology-specific pre-trained models. This demonstrated that, while keeping the model structure fixed, domain-specific knowledge in the image encoder and LLM can significantly enhance the model's performance.
>
> To present a more rigorous ablation study, we will incorporate your suggestions in the revised manuscript while retaining the previous ablations to ensure a more comprehensive and fair comparison.
>
> *[g] Pérez-García, Fernando, et al. "RAD-DINO: Exploring Scalable Medical Image Encoders Beyond Text Supervision." arXiv preprint arXiv:2401.10815 (2024).*
>
> *[h] Bosselut, Antoine, et al. "MEDITRON: Open Medical Foundation Models Adapted for Clinical Practice." (2024).*
>
> ***
> **W4. Clarity in Table 5: Table 5 lacks clarity. Notably, using only Findings in Stage 1 yields better performance in Clinical Efficacy. One would intuitively expect the VQA dataset, which emphasizes detailed symptom descriptions, to enhance Clinical Efficacy, yet the results in Table 3 show a decrease.**
>
> **A4.**
>
> Thanks for your constructive suggestions. In the revised manuscript, we will clarify this point to address any ambiguity. First, it is **not** possible to fairly compare the performance of these two models on the VQA dataset. Additionally, our focus is on MLLMs for the RRG task, and testing on VQA datasets is beyond our study scope. **(Table 3** shows an **increase**; as discussed in **"A-2"**, the model's performance is better with true prior images.**)**
>
> At the beginning of **Section 4 (line 384)**, we mention that all ablation models were tested on the MIMIC-CXR official test split for the findings section generation task. Meanwhile, in **line 448**, we note the "Only Findings" model as a comparison model (in **Table 5**). This model shares the same structure as Libra but uses only partial data for alignment in the first training phase, while the second phase uses the same dataset. Both phases are evaluated on the findings section generation task.
>
> Meanwhile, in **Section 4 (Q3, Table 5)**, the comparison models were not trained on VQA datasets, making fair comparison **infeasible** on the VQA datasets.
>
> * In the first training stage, the Libra model uses additional datasets to train the TAC, while "Only Findings" focuses solely on learning the data distribution of the findings section dataset. Libra's first-phase alignment involves learning a substantially larger data distribution, which makes it **less** specialised for the RRG task. VQA datasets indeed emphasize detailed descriptions of a single symptom for each scan, such as "Is it pleural effusion on the scan? - Yes"
>
> However, as mentioned earlier, our evaluation is based on **report generation**, where a report typically encompasses multiple symptoms, not a single one. In **Table 1 (line 275)**, we report that tasks for the findings section generation account for only **13%** of the entire dataset. Consequently, after being trained on such a dataset, Libra is not expected to excel on this single-symptom task following the first training stage.
>
> * In the second training stage, both Libra and "Only Findings" undergo LLM fine-tuning on the same dataset to further optimise text output quality. In **Section 2.4 (line 240)**, we mention that during this phase, the TAC remains frozen, meaning the learned image representations during the alignment phase remain unchanged. Libra's TAC, having learned a richer representation than "Only Findings" in the alignment phase. Consequently, Libra performs better in the second evaluation.

---

> > ### Comment · Reviewer_y9E4 · 2024-11-24
> >
> > For *W3*, please show the results of (RAD-DINO + Meditron without any extra modules). If you have presented the results in Appendix B.1, highlight the row.

---

> > > ### Author Response · Authors · 2024-11-24
> > >
> > > Dear Reviewer y9E4,
> > >
> > >
> > >
> > >
> > > I just uploaded the latest manuscript; however, due to the large size of the table, it cannot be directly included here.
> > >
> > >
> > > You can find the results highlighted in red in **Section 4 (second ablation, line 408)**, showcasing experiments conducted with settings consistent with Libra's architecture (i.e., RAD-DINO and Meditron).
> > >
> > >
> > > Additionally, the ablation study using general-domain pre-trained models (i.e., DINOv2 and Vicuna-7B) has been relocated to **Appendix C.1 (line 1358)** for better organisation.
> > >
> > >
> > >
> > >
> > > Best regards,
> > >
> > >
> > > Authors

---

> > > > ### Comment · Reviewer_y9E4 · 2024-11-25
> > > >
> > > > Thank you for updating your paper. There may be a typo in line 420 of the revised paper. It seems that only Stage 1 is conducted for ablation studies in Table 4. If possible, providing results with stage 2 would be better.  If my queries are answered with sufficient detail, I am open to re-evaluating my decision.

---

> > > > > ### Author Response · Authors · 2024-11-25
> > > > >
> > > > > Dear Reviewer y9E4,
> > > > >
> > > > >
> > > > > Thanks for your attentive feedback. In the latest manuscript, we have corrected these typos.
> > > > >
> > > > > In **Appendix C.2 (Table 9, line 1404)**, we provide the ablation study results with the stage 2 training.
> > > > >
> > > > > We included this ablation study in the appendix for the following reasons:
> > > > >
> > > > > * In the stage 2 ablation study, two variables are considered: the weights within the **Temporal Alignment Connector** (TAC) and the **LoRA** weights. While TAC and its ablated variants remain frozen during fine-tuning, variations in TAC weights can still influence the training of LoRA weights in the LLM.
> > > > >
> > > > > * This ablation conflicts somewhat with the principle of the **control variable**. Ideally, the control group and treatment group should differ by only **one** variable to isolate the effect of the variable being studied.
> > > > >
> > > > > However, the conclusion of this ablation study remains consistent with **Section 4 (Q2, line 433)**, demonstrating that the performance improvements brought by TAC are stable and unaffected by changes in training stages.
> > > > >
> > > > > ***
> > > > >
> > > > > Additionally, we included another intriguing ablation in **Appendix C.4 (Table 10)**. This experiment involves **reinitialising** a **new LoRA adapter** for the model after the stage 2 training and performing another round of fine-tuning. While this approach induces overfitting, it further demonstrates that the training strategy and stage do not affect the performance improvements brought by TAC.
> > > > >
> > > > > As noted in **Appendix C.4 (line 1469)**, TAC reliably embeds the capacity for temporal information processing into our model, making it a stable and essential component for handling temporal data in RRG tasks. Simply put, the impact of TAC can be described as **"permanent deformation"**.
> > > > >
> > > > >
> > > > > We appreciate your suggestions regarding the ablation study, which will make it stronger!
> > > > >
> > > > >
> > > > >
> > > > > Best regards,
> > > > >
> > > > >
> > > > > Authors

---

> > > > > > ### Comment · Reviewer_y9E4 · 2024-11-26
> > > > > >
> > > > > > Given the details provided in the response, I have raised my score.

---

> > > > > > > ### Author Response · Authors · 2024-11-26
> > > > > > >
> > > > > > > Dear Reviewer y9E4,
> > > > > > >
> > > > > > >
> > > > > > > We sincerely appreciate your thoughtful consideration of our responses and for raising your score. Your feedback has been instrumental in refining our work. If you have any additional suggestions or concerns, we would be more than happy to address them to further enhance the quality of our manuscript.
> > > > > > >
> > > > > > >
> > > > > > > Thank you once again for your support and encouragement.
> > > > > > >
> > > > > > >
> > > > > > > Best regards,
> > > > > > >
> > > > > > >
> > > > > > > Authors

---

> > ### Comment · Reviewer_y9E4 · 2024-11-24
> >
> > Can you explain the reason why pretraining on *Only findings* during Stage 1 shows better results in the clinical efficacy of Table 5?

---

> > > ### Comment · Reviewer_y9E4 · 2024-11-24
> > >
> > > If possible, please present detailed examples of prompts (including clinical context in **Section 2.3**) for generating reports.

---

> > > ### Author Response · Authors · 2024-11-24
> > >
> > > Dear Reviewer y9E4,
> > >
> > >
> > > Thank you very much for your suggestion. In the latest manuscript, we have added detailed examples of prompts, which can be found in **Appendix B.4 (line 1298)**.
> > >
> > >
> > >
> > >
> > > Below is an example to illustrate the prompt design. Meanwhile, a radiology report is provided in **Appendix B.4 (Table 7)** to help readers better understand the prompt structure.
> > >
> > >
> > >
> > >
> > > ***
> > > **[System prompt] {** The assistant specialised in comparing Chest X-ray images, identifying differences, and noting temporal changes.**}** \+
> > >
> > >
> > > *<Image Representation Placeholder>* \+
> > >
> > >
> > > **[Clinical prompt] {** Provide a detailed description of the findings in the radiology image. Following clinical context:
> > >
> > >
> > > *Indication*: Dyspnea and cough, right-sided back pain. *History*: Intubation with pulmonary edema. *Comparison*: Chest radiographs on \_\_\_ and CT chest without contrast on \_\_\_. *Technique*: Portable upright chest radiograph. **}**
> > >
> > >
> > >
> > >
> > > ***
> > > Best regards,
> > >
> > >
> > > Authors

---

> > > ### Author Response · Authors · 2024-11-24
> > >
> > > Dear Reviewer y9E4,
> > >
> > >
> > > Thank you for your follow-up question. We apologise if the explanation of **W4** was not sufficiently clear. Below, we provide an example for clarification.
> > >
> > >
> > > * In the first training phase, **Libra** trains the adapter on a larger dataset. During this phase, the model is trained on both the report generation task and the VQA task. The whole dataset is shuffled and used for holistic training.
> > >
> > >
> > > * The **Only findings** model is trained exclusively on data specific to the report generation task.
> > >
> > >
> > > This end-to-end training approach enables **Libra** to learn the distribution of samples across the entire dataset, thereby achieving better generalisation. As mentioned in **A4**, the VQA data constitutes 87% of the training dataset.
> > >
> > >
> > > ***
> > >
> > >
> > > > VQA dataset consists of questions like:
> > > * Question: "Is there pleural effusion on the scan?"
> > >   * Answer: "Yes."
> > >
> > >
> > > Here, the answers are typically very short, as the dataset focuses on identifying a **single** symptom in the scan. On average, answers in the VQA dataset contain approximately 10 tokens.
> > >
> > >
> > > ***
> > > > Report generation dataset:
> > > * Question: “Provide a detailed description of the findings in the radiology image.”
> > >   * Answer: "In comparison with the prior study, there are diffuse bilateral pulmonary opacifications, more prominent on the right. Severe pulmonary edema, but pneumonia or development of ARDS cannot be ruled out. Monitoring and support devices are appropriately positioned."
> > >
> > >
> > > Here, the answers are significantly longer, ranging from 25 to 150 tokens, as they describe **multiple** symptoms.
> > > ***
> > >
> > >
> > > In autoregressive models, early stopping serves as a regularisation method to prevent overfitting, enabling the model to balance learning the dataset's distribution. During inferencing, the early stopping mechanism impacts the **output length** generated by the two models.
> > >
> > >
> > > ***
> > > > Generated Answers:
> > > * **Libra**: "In comparison with the prior study, there are diffuse bilateral pulmonary opacifications, more prominent on the right."
> > > * **Only findings model**: "In comparison with the prior study, indicate severe pulmonary edema. Diffuse bilateral pulmonary opacifications, more prominent on the right. The devices are appropriately positioned."
> > > ***
> > > As shown above, Libra generates shorter outputs, focusing on single symptoms even in the report generation task. After the first stage, **Libra** is *better* at VQA task. In contrast, the **Only findings** model produces more detailed descriptions, as its training prioritises generating comprehensive content.
> > >
> > >
> > > This demonstrates that the **Only findings** model achieves **better results in clinical metrics** but performs a **decrease in lexical metrics**.
> > >
> > >
> > > * This inconsistency is linked to the structured nature of radiology reports, where symptom descriptions are discrete and lack connections between sentences.
> > >
> > >
> > > Due to differences in **sentence order**, metrics evaluating lexical fluency will decline, even when the content accurately describes multiple symptoms.
> > >
> > > ***
> > >
> > >
> > > Best regards,
> > >
> > >
> > > Authors

---

> ### Author Response · Authors · 2024-11-20
> **Response to Reviewer y9E4 (4/4)**
>
> (Follow-Up to **A4**)
>
> This setup aligns with most previous work in MLLMs' training, where the first phase involves using a much larger dataset for image-text alignment, while the second phase utilises a relatively smaller dataset for fine-tuning downstream tasks. Thanks again for your suggestion. We will revise the model name "**Only Findings**" (in Table 5) to "**Model (Only Findings)**" to improve the clarity of the latest manuscript.
>
> ***
> **W5. Explanation of Figure 5: Provide more detailed explanations for Figure 5 in the Appendix, where the heatmap indicates that the Prior LFE shows more focused attention.**
>
> **A5.**
>
> **Figure 5** is **not** intended to demonstrate that "Prior LFE shows more focused attention." We will provide a detailed explanation in the revised manuscript to reduce ambiguity from the heatmap.
>
> The caption of **Figure 5 (Appendix D)** mentions the heatmap is the visualisation of the *'output_hidden_states'* from the TAC module after image input for the RRG task. This differs from traditional phrase grounding tasks. As noted in **Section 2.3 (line 224)**, the prompts we used do not include phrases with specific symptoms but are solely designed for instruction tuning MLLMs for the RRG task.
>
> * **Figure 5** highlights the change in the attention map of the TAC's output when the current and prior images are swapped. The representations from LFE remain **unchanged** despite the swap, indicating that LFE's output does not incorporate temporal information. After swapping the order, the hidden layer outputs of the TAC module **changed**, as shown in Figure 5, where TAC-1 and TAC-2 exhibit noticeable differences.
>
> This result is expected, as the TAC's residual connection ensures that the current image remains the main modality while the prior image serves as the auxiliary modality. When the order is swapped, the prior image becomes the main modality for subsequent processing.
>
> * Swapping the order of images also alters the content of the generated report, as mentioned in **Section 5 (line 518)**. The primary difference lies in the **reversal of the temporal state of symptoms**, such as changing from "improving" to "worsening." The reference image serves as the starting point for event observation, while the current image represents the endpoint. Since the same prompt is used, the variation in the output report arises from the temporal information representations generated by the two images through the TAC module.
>
> We sincerely appreciate your interesting suggestion. We also acknowledge that the prior image in the top right corner of **Figure 5** inadvertently displayed a black border during the LaTeX formatting process. We will correct this issue in the revised manuscript.
>
> ***
> We sincerely hope that our responses have addressed your concerns and hope you will consider increasing your score. If we have left any notable points of concern overlooked, we would greatly appreciate your feedback, and we will attend to these points. Additionally, we will incorporate all the suggestions and discussions mentioned above in the latest manuscript. Thanks again for your thoughtful review and consideration.

---

### Official Review · Reviewer_Y2it · 2024-11-02

**Soundness:** 2
**Presentation:** 3
**Contribution:** 2
**Rating:** 5
**Confidence:** 3

**Summary:**

This paper introduces Libra, a vision-language model designed for radiology report generation based on chest X-rays. Libra integrates temporal information across images taken at different time points through a Temporal Alignment Connector (TAC), enabling the model to capture temporal changes effectively. The model leverages a pre-trained RAD-DINO image encoder and a specialized medical language model, Meditron, to generate high-quality radiology reports. Experiments demonstrate that Libra outperforms existing VLM methods on the MIMIC-CXR dataset.

**Strengths:**

* Innovative temporal processing. The TAC module is a novel addition that allows Libra to capture and utilize temporal changes in medical images effectively, enhancing the model's clinical applicability.
* Comprehensive ablation studies. The ablation experiments clarify the importance of each submodule (TFM, LFE, and PIPB), reinforcing the credibility of the design choices.
* Comprehensive appendix. The appendix is highly commendable, providing detailed descriptions of the datasets, training configurations, and additional experimental results. This thorough documentation enhances the paper’s transparency and reproducibility, offering valuable insights that complement the main text and support further exploration by readers.

**Weaknesses:**

Despite the paper's clarity, several imprecise arguments and overstatements necessitate revision and clarification:
* Incomplete framework representation. TFM is a crucial component of the core TAC, but the framework diagram omits the illustration of the $MLP_{final}$ part within TFM. This omission may lead to ambiguity regarding the final processing steps, making it more difficult for readers to fully understand how all modules are integrated within the model. It is recommended that the authors update the framework diagram to include this component and provide a detailed explanation of its role, which would enhance the overall clarity of the model architecture.
* Insufficient detail on RAD-DINO. The description of RAD-DINO in the model architecture section is too brief, lacking the depth necessary for readers to fully understand its role and integration within the model. It is suggested that the authors provide a more comprehensive explanation, including details about its architecture, pre-training process, and specific contributions to the model’s performance in radiology tasks. This would improve both the clarity and the completeness of the proposed framework.
* Over-reliance on prior images. Although the paper introduces the Prior Image Prefix Bias to address the absence of prior images, the model still faces challenges during generation when prior images are unavailable. This suggests that the current solution may not fully resolve issues related to temporal information gaps, potentially affecting the model’s robustness in real-world clinical scenarios. The authors are encouraged to conduct further analysis of the model’s performance without prior images and explore potential strategies to enhance robustness under limited temporal information.
* Lack of clear novelty in vision-language alignment. The proposed vision-language alignment mechanism lacks distinct novelty when compared to existing methodologies in this domain. Similar cross-modal attention mechanisms have been extensively explored in prior studies. The authors are advised to conduct a detailed comparison between their approach and previous works, specifically highlighting any improvements or innovations that set their model apart in the context of radiology report generation tasks.
* Insufficient acknowledgment of prior work. The paper lacks a thorough review of relevant prior work, especially recent advancements in vision-language models for medical image interpretation. This omission makes it challenging to contextualize the model's contributions. The authors should provide an in-depth discussion of related works to clearly establish how Libra advances the state of the art and addresses specific gaps in the literature.
* 	Limited dataset and evaluation metrics. The experimental evaluation relies solely on the MIMIC-CXR dataset, which raises concerns about the model's generalizability. Additionally, some evaluation metrics are insufficiently explained. It is recommended that the authors include experiments on other datasets, such as CheXpert, and offer more detailed explanations of the evaluation metrics to strengthen the model's validation and applicability.

**Questions:**

Generalizability. How would the model perform on other datasets or in different clinical settings?

**Details Of Ethics Concerns:**

The required dataset includes patient medical records, which may raise ethical concerns related to patient privacy.

---

> ### Author Response · Authors · 2024-11-20
> **Response to Reviewer Y2it (1/6)**
>
> Dear Reviewer Y2it,
>
> Thank you for your valuable feedback and constructive suggestions. We sincerely appreciate your recognition of our innovative temporal processing and comprehensive ablation studies. Our detailed response to your concerns is listed as follows:
> ***
>
> **W1. Incomplete framework representation. TFM is a crucial component of the core TAC, but the framework diagram omits the illustration of the $MLP_{final}$ part within TFM. This omission may lead to ambiguity regarding the final processing steps, making it more difficult for readers to fully understand how all modules are integrated within the model. It is recommended that the authors update the framework diagram to include this component and provide a detailed explanation of its role, which would enhance the overall clarity of the model architecture.**
>
> **A1.**
> Our framework is complete. In **Figure 1**, the framework diagram does **not** omit the illustration of $MLP_{final}$, which is displayed as a **larger** blue Multi-Layer Perceptron module. We greatly appreciate your suggestions to improve the readability and clarity of our manuscript, which will make it stronger!
>
> We will update the framework diagram by using different colours to distinguish between $MLP_{attn}$ and $MLP_{final}$ and revise *$Equation (7)$* (in **line 210**) to reduce ambiguities and enhance the overall clarity of the model architecture. The updates are as follows:
>
> >  $$
> Z_{img}=TFM(A\^{curr}\_{img},A\^{prior}\_{img})  \iff Z_{img} = MLP_{final}(T^{out}_{img}) \tag{7}
> $$
>
> Additionally, in **Section 2.2.2 (line 205)**, *$Equation (6)$* shows that $MLP_{attn}$ is followed by a residual connection and layer normalization, whereas *$Equation (7)$* shows that $MLP_{final}$ does **not** include these operations. And $Z_{img}$ is the output of Temporal Alignment Connector (TAC), serving as the feature vector aligned with the LLM. We will revise the manuscript based on the discussions above.

---

> ### Author Response · Authors · 2024-11-20
> **Response to Reviewer Y2it (2/6)**
>
> ***
> **W2. Insufficient detail on RAD-DINO. The description of RAD-DINO in the model architecture section is too brief, lacking the depth necessary for readers to fully understand its role and integration within the model. It is suggested that the authors provide a more comprehensive explanation, including details about its architecture, pre-training process, and specific contributions to the model’s performance in radiology tasks. This would improve both the clarity and the completeness of the proposed framework.**
>
>
> **A2.**
>
> **RAD-DINO** [a] is a powerful mode that has been widely used in the radiology report generation (RRG) research, including MAIRA-2 [d], M4CXR [e] and [a], therefore we omit the details of RAD-DINO in our current manuscript. However, we will provide a more detailed description of RAD-DINO as the appendix of the final version of the paper to make it more self-contained.
>
> In **line 107**, we highlight that RAD-DINO is continually pre-trained with medical scans using the DINOv2 [b] framework, a classic image-only self-supervised model. Since this is already a pre-trained model, no further fine-tuning or continued training is required. Meanwhile, in **Section 2.2.1 (line 153)**, we outline the use of RAD-DINO, including the number of layers, the process of generating patch embeddings, and how we leverage these embeddings for downstream tasks. As stated in **Section 2.4 (line 237)**, the image encoder remains frozen throughout the entire training process. Therefore, a detailed explanation of the pre-training process for RAD-DINO is beyond the scope of our study.
>
> Additionally, we compared the performance of several image encoders (**RAD-DINO** [a], **DINOv2** [b], and **CLIP** [c]), and the results show that RAD-DINO achieved better performance.
>
> * In **Appendix B.1**, we examine the effect of replacing the image encoder with a general-domain pre-trained model (DINOv2), highlighting the importance of domain-specific knowledge and justifying our choice of RAD-DINO.
> * In **Appendix B.2**, we initialise another type of pre-trained model (CLIP), which applies contrastive learning to both language and images.
>
> Both DINOv2 and CLIP represent classic architectures for pre-training vision transformers (ViTs). For other classical models, such as ResNet50-based image encoders, their performance on RRG tasks is limited, as validated in **[a]**, falling short of ViT-based. Therefore, a detailed comparison of different types of pre-trained image encoders is outside the scope of our study.
>
> In **line 68**, we mention that current multimodal large language models (MLLMs) generally rely on a single hidden state layer from the pre-trained image encoder. In contrast, our model extracts features from all hidden layers and fuses them through the Temporal Alignment Connector, enabling the integration of image features with temporal information while aligning them with the dimensions of the LLM. Therefore, our study **focuses on** leveraging and aligning temporal images within MLLMs on the RRG task.
>
> We sincerely appreciate your kind suggestion. In the revised manuscript, we will consolidate the above ablation studies into a dedicated section to thoroughly explore several different pre-trained image encoders and clearly compare their effects on Libra. This will help readers gain a more comprehensive understanding of their role and integration within the framework.
>
> *[a] Pérez-García, Fernando, et al. "RAD-DINO: Exploring Scalable Medical Image Encoders Beyond Text Supervision." arXiv preprint arXiv:2401.10815 (2024).*
>
> *[b] Oquab, Maxime, et al. "Dinov2: Learning robust visual features without supervision." arXiv preprint arXiv:2304.07193 (2023).*
>
> *[c] Radford, Alec, et al. "Learning transferable visual models from natural language supervision." International conference on machine learning. PMLR, 2021.*
>
> *[d] Bannur, Shruthi, et al. "Maira-2: Grounded radiology report generation." arXiv preprint arXiv:2406.04449 (2024).*
>
> *[e] Park, Jonggwon, et al. "M4CXR: Exploring Multi-task Potentials of Multi-modal Large Language Models for Chest X-ray Interpretation." arXiv preprint arXiv:2408.16213 (2024).*

---

> ### Author Response · Authors · 2024-11-20
> **Response to Reviewer Y2it (3/6)**
>
> ***
> **W3. Over-reliance on prior images. Although the paper introduces the Prior Image Prefix Bias to address the absence of prior images, the model still faces challenges during generation when prior images are unavailable. This suggests that the current solution may not fully resolve issues related to temporal information gaps, potentially affecting the model’s robustness in real-world clinical scenarios. The authors are encouraged to conduct further analysis of the model’s performance without prior images and explore potential strategies to enhance robustness under limited temporal information.**
>
> > **Review:** Over-reliance on prior images.
>
> **A3 (Answer-1).**
>
> In terms of methodology, we use multiple residual connections to link the features of the current image with its intermediate output  (see **Section 2.2.2**, *$Equation (3-6)$*), ensuring there is **no over-reliance** on prior images.
>
> > **Review:** Although the paper introduces the Prior Image Prefix Bias to address the absence of prior images, the model still faces challenges during generation when prior images are unavailable.
>
> **A3 (Answer-2).**
>
> To be clarified, the Prior Image Prefix Bias (PIPB) is **not** designed to address the absence of prior images.
>
> In **Section 2.2.2 (line 184)**, we define that if no prior image is available, the current image is used as a dummy prior image. As shown in **Figure 1**, our model consistently processes two images: one as the current image and the other as either the prior or dummy prior image. This setup ensures that the model **never encounters** cases when prior images are unavailable.
>
> In summary, PIPB is designed to enhance the model's robustness, while the use of a dummy prior image is designed to address the absence of prior images.
>
> * PIPB is better understood as a mechanism for distinguishing the temporal information between the current image and the prior image, especially the dummy prior image. As shown in **Figure 1**, **without** PIPB, the Temporal Fusion Module (TFM) would actually apply self-attention to the current image twice, as the Q, K, and V would always be same. This causes both sides of the Self-Attention block to explore the same latent space. As a result, the Q, K, and V remain identical in subsequent Cross-Attention block, leading to redundant iterations.
> * These iterations increase depth but fail to capture meaningful information (i.e.,  temporal changes in short intervals). PIPB serves as a temporal information "noise" added to the dummy prior image, allowing Cross-Attention to effectively detect such extreme cases and enhancing the model's robustness.
>
>
> > **Review:** This suggests that the current solution may not fully resolve issues related to temporal information gaps, potentially affecting the model’s robustness in real-world clinical scenarios.
>
> **A3 (Answer-3).**
>
> Our approach could align with real-world clinical scenarios. The **temporal information** is derived from comparisons between scans taken at different time points. In practice, clinicians rely on the latest scan in combination with prior images to inform their diagnoses. These prior images may include scans from multiple time points, and referencing scans taken at different intervals can result in varying descriptions of symptom changes. Such differences in comparisons are ultimately reflected in the diagnostic report.
>
> * For example, in a specific case where a patient has two scans just milliseconds apart, the current and prior images would be nearly identical, as no pathological changes would manifest within such **a short interval**. This extreme scenario simulates how the model handles clinical practice under limited temporal information.
>
> In such case, the correct diagnosis for this tiny time interval would be that the patient's condition is **"stable"**. Thus, the dummy prior image addresses scenarios where a true prior image is absent. This approach improves the model's robustness by training it to correctly interpret temporal information even in cases with very short time intervals.

---

> ### Author Response · Authors · 2024-11-20
> **Response to Reviewer Y2it (4/6)**
>
> (Follow-Up to **A3**)
>
> > **Review:** The authors are encouraged to conduct further analysis of the model’s performance without prior images and explore potential strategies to enhance robustness under limited temporal information.
>
> **A3 (Answer-4).**
>
> In the first ablation study of **Section 4**, we conducted comparative experiments during inference to analyse the model's performance without true prior image. In our study, without prior images does **not entirely equate** to without temporal information; rather, it represents a scenario of **limited temporal information**, as you mentioned.
>
> Therefore, when only the dummy prior image is used, the model explores its performance under conditions without prior images. In **Section 2**, we evaluated the performance across the entire test split. In **Appendix B.2 (line 1057)**, we separately evaluated samples with and without prior images in the test split but used only a single evaluation metric.
>
> Throughout end-to-end training, the model has learned to interpret various temporal information related to symptoms, such as "stable," "worsening," or "improving."
> * Our model does **not** rely on a symbolic way to indicate the presence or absence of a prior image. Instead, we treat 'absent' as a unique temporal information scenario, using a dummy prior image.
>
> We sincerely appreciate your kind suggestion. In the revised manuscript, we will include a **Limitations and Future Work** section with further analysis of complex scenarios involving specific temporal information (e.g., handling multiple prior images).
>
> ***
> **W4. Lack of clear novelty in vision-language alignment. The proposed vision-language alignment mechanism lacks distinct novelty when compared to existing methodologies in this domain. Similar cross-modal attention mechanisms have been extensively explored in prior studies. The authors are advised to conduct a detailed comparison between their approach and previous works, specifically highlighting any improvements or innovations that set their model apart in the context of radiology report generation tasks.**
>
> **A4.**
>
> We sincerely appreciate your suggestions to enhance the novelty of our manuscript, which will make it stronger!
>
> Our model differs from existing methods in terms of vision-language alignment for temporal information on RRG. As noted in **line 60**, current parallel works employ the adapter that only accept a single image as input. Meanwhile, these adapters rely on a single hidden layer representation from the pre-trained encoder (either the last or penultimate layer) and cannot integrate information from multiple images simultaneously. The way these MLLMs handle multiple image inputs is by **concatenating** multiple images and text as a combined input to the LLM, for example, *"<image 1 placeholder>+<image 2 placeholder>+<prompt>"*.
>
> * In contrast, our model, as described in **Section 2.2.1**, leverages all hidden layer features from the image encoder, which distinguishes it from previous works. This approach allows our model to use these multi-layered features for higher granularity in feature representation.
> * Additionally, our Temporal Alignment Connector (TAC) consistently receives two images as input. As shown in **Section 2.2.2**, *$Equation (3-6)$*, the residual connections allow the prior image or dummy prior image to serve as an auxiliary modality, complementing the current image. In end-to-end training, the model aligns not only image and text representations but also incorporates temporal information, setting it apart from existing methods.
>
> Meanwhile, to ensure a rigorous statement, we will replace **"visual language models (VLMs)"** with **"multimodal large language models (MLLMs)"** in the revised manuscript. We acknowledge that VLMs broadly cover multimodal models that learn from images and text, including large language models (LLMs) based models, transformer encoder-decoder models, and their variants. As LLM-based models typically outperform other types of models on RRG tasks, we primarily compare Libra with other LLM-based models to ensure a fair evaluation.
>
> In the revised manuscript, we will include a detailed discussion of prior work to more clearly highlight the innovations our model brings to the RRG task.

---

> ### Author Response · Authors · 2024-11-20
> **Response to Reviewer Y2it (5/6)**
>
> ***
> **W5. Insufficient acknowledgment of prior work. The paper lacks a thorough review of relevant prior work, especially recent advancements in vision-language models for medical image interpretation. This omission makes it challenging to contextualize the model's contributions. The authors should provide an in-depth discussion of related works to clearly establish how Libra advances the state of the art and addresses specific gaps in the literature.**
>
> **A5:**
>
> We acknowledge that our explanation of prior work on RRG is insufficient. In **lines 59-67**, we provided only a brief introduction to existing parallel work.
>
> In the latest manuscript, we will add a detailed discussion and review of relevant prior works on the RRG task. Based on the discussion in **"A-4"**, we will provide a more extensive analysis of prior studies.
>
> * **LLM-based** models are typically trained in an end-to-end manner. In contrast, **non-LLM-based** models use a different architecture, generally comprising multiple modules that require separate training.
> * Meanwhile, non-LLM-based models handle single and double image inputs by symbolically differentiating tasks and employing two separate neural network architectures: one for single-image inputs and another for double-image inputs.
>
> Given these differences, non-LLM-based models are typically not compared with those of LLM-based models in their paper. Nonetheless, we will include a discussion of all these models to provide readers with a broader understanding of the latest advancements in vision-language models for the RRG task. We welcome any suggestions for relevant works we may have missed.
>
> ***
> **W6. Limited dataset and evaluation metrics. The experimental evaluation relies solely on the MIMIC-CXR dataset, which raises concerns about the model's generalizability. Additionally, some evaluation metrics are insufficiently explained. It is recommended that the authors include experiments on other datasets, such as CheXpert, and offer more detailed explanations of the evaluation metrics to strengthen the model's validation and applicability.**
>
> **A6 (About dataset).**
>
> We greatly appreciate your constructive suggestions regarding the model's generalizability. Our study selected the most suitable and reliable dataset. **MIMIC-CXR** [f] is not only publicly accessible but also provides complete ground truth labels for radiology scans, including scan time, view type, and full diagnostic reports.
>
> We thoroughly investigated other datasets but found they were unsuitable for our study. Key considerations in our dataset selection were the presence of complete reports and the availability of temporal information (i.e., prior images).
>
> * **CheXpert** [g] provides annotated scans with label-specific annotations rather than full medical reports, making it better suited for training image encoders or annotation models focused on automated chest X-ray interpretation. Therefore, it is not appropriate for the RRG task.
> * **PadChest** [h] provides reports and corresponding prior images, but it is in Spanish, placing cross-language training beyond the scope of our study.
> * The **IU-Xray** [i] dataset lacks patient-level metadata and prior study information, which contradicts our focus on temporal information in images.
> * Additionally, MIMIC-CXR derivatives, such as **Chest ImaGenome** [j], do not adhere to the official data split. Using these datasets would risk data leakage across train, valid, and test.
>
> Considering the above points, and as mentioned in **Appendix A.2**, to strictly prevent data leakage, we ensured that all datasets adhered to the official split in MIMIC-CXR. This setup guarantees fair model comparisons between our study and other models.
>
> **A6 (About metrics).**
>
> We also appreciate your constructive suggestions regarding the limited evaluation metrics.
>
> As described in **Appendix A.3**, we followed prior works and used as many evaluation metrics as possible to ensure a comprehensive assessment. In **line 921**, we indeed introduced the lexical and radiology-specific metrics used, along with their setup, to facilitate fair comparisons.
>
> In the revised manuscript, we will provide more detailed discussions and explanations about other datasets, as well as evaluation metrics.
>
> *[f] Johnson, Alistair EW, et al. "MIMIC-CXR, a de-identified publicly available database of chest radiographs with free-text reports."*
>
> *[g] Irvin, Jeremy, et al. "Chexpert: A large chest radiograph dataset with uncertainty labels and expert comparison."*
>
> *[h] Bustos, Aurelia, et al. "Padchest: A large chest x-ray image dataset with multi-label annotated reports."*
>
> *[i] Demner-Fushman, Dina, et al. "Preparing a collection of radiology examinations for distribution and retrieval."*
>
> *[j] Wu, Joy T., et al. "Chest imagenome dataset for clinical reasoning."*

---

> ### Author Response · Authors · 2024-11-20
> **Response to Reviewer Y2it (6/6)**
>
> ***
> **Q1. Generalizability. How would the model perform on other datasets or in different clinical settings?**
>
> **A-Q1.**
>
> In the revised manuscript, we will claim this interesting suggestion regarding the model's generalisability. Specifically, we will discuss the challenges and limitations of our model on other datasets and in different clinical settings, to better outline future research directions.
>
> * It is important to note that the research object in our study is the **chest X-ray** (frontal view), not the patient. We selected the most suitable dataset based on our research object, as answered in **"A-6".** Our target task is RRG, where the clinical setting is **fixed**: given a scan and a prior scan (if available), the goal is to generate the corresponding report section about the primary scan.
>
> For fairness, our model was trained exclusively on MIMIC-CXR at all stages, including alignment and fine-tuning. Testing was also conducted solely on this dataset to enable fair and direct comparisons with other LLM-based models. Other datasets were not used for training, as they would not provide a fair basis for comparison. Scenarios involving other clinical settings, such as lateral view scans or multiple current/prior scans, fall beyond the scope of our study.
>
> In the revised manuscript, we will add a **Limitations and Future Work** section to further discuss the model's performance in different clinical settings. We welcome any specific suggestions regarding the challenges faced by the model.
>
> ***
> **Details Of Ethics Concerns: The required dataset includes patient medical records, which may raise ethical concerns related to patient privacy.**
>
> **Answer.**
>
> Thank you for highlighting this point.
> * MIMIC-CXR is a publicly available dataset, and the official statement confirms that it is "de-identified to satisfy the US Health Insurance Portability and Accountability Act of 1996 (HIPAA) Safe Harbor requirements. Protected health information (PHI) has been removed. The dataset is intended to support a wide body of research in medicine, including image understanding, natural language processing, and decision support.”
> * Furthermore, "the collection of patient information and creation of the research resource was reviewed by the Institutional Review Board at the Beth Israel Deaconess Medical Center, who granted a waiver of informed consent and approved the data sharing initiative."
>
> Our research and experiments are conducted in compliance with these terms.
>
> ***
> We sincerely hope that our responses have addressed your concerns and hope you will consider increasing your score. If we have left any notable points of concern overlooked, we would greatly appreciate your feedback, and we will attend to these points. Additionally, we will incorporate all the suggestions and discussions mentioned above in the revised manuscript. Thanks again for your thoughtful review and consideration.

---

> ### Comment · Reviewer_Y2it · 2024-11-24
> **Thank you for the responses.**
>
> Thank you for your detailed response to the review comments and the further clarification. Your explanations are comprehensive and well-articulated, effectively addressing the concerns raised. I have no further questions and maintain my original rating.

---

> ### Author Response · Authors · 2024-11-25
>
> Dear Reviewer Y2it,
>
>
> Thank you very much for taking the time to review our paper. We also sincerely appreciate your acknowledgement of our explanations.
>
> If you still need any clarification or have any other concerns, please feel free to contact us and we are happy to continue communicating with you.
>
> Best regards,
>
> Authors

---

### Official Review · Reviewer_poux · 2024-11-04

**Soundness:** 2
**Presentation:** 3
**Contribution:** 2
**Rating:** 5
**Confidence:** 4

**Summary:**

The authors propose a method that generates chest X-ray reports from temporal inputs. The method combines a radiology-specific image encoder with a VLM and introduces a Temporal Alignment Connector (TAC) to capture and synthesize temporal information from images taken at different times. Experiments demonstrate that the method achieves good performance among other vision language models on the MIMIC-CXR dataset.

**Strengths:**

1- The clinical problem statement is fair and important
2- The evaluation is good and comprehensive and ablation studies showed how the method behaves in difference scenarios
3- The authors introduced an interesting technical local and global learning mechanism

**Weaknesses:**

1- The main claim of the paper is that it is the first to introduce a VLM for automatic report generation that utilizes temporal scans to ensure more realistic reports that learn from multiple scans acquired at different time points. Regardless of whether it is a VLM or other types of encoder/decoder nets, this claim is not true because multiple works have been published to address this problem. For instance,

-https://aclanthology.org/2023.findings-emnlp.325/
-https://aclanthology.org/2023.findings-emnlp.140/
- https://arxiv.org/abs/2403.13343
- https://arxiv.org/pdf/2301.04558 (which you already cite but don't compare with).

2- It is not clear whether the method is capable of handling one previous scan or multiple

3- In Figure 1, the effect of Rad-Dino is not clear. What if it is replaced with a classical pre-trained image encoder?

**Questions:**

Please respond to the three points in the weaknesses.

---

> ### Author Response · Authors · 2024-11-20
> **Response to Reviewer poux (1/3)**
>
> Dear Reviewer poux,
>
> Thank you for your valuable feedback and constructive suggestions. We sincerely appreciate your acknowledgement of our ablation studies across different scenarios. Our detailed response to your concerns is listed as follows:
> ***
> **W1. The main claim of the paper is that it is the first to introduce a VLM for automatic report generation that utilizes temporal scans to ensure more realistic reports that learn from multiple scans acquired at different time points. Regardless of whether it is a VLM or other types of encoder/decoder nets, this claim is not true because multiple works have been published to address this problem.**
>
> **A1.**
>
> We sincerely appreciate your suggestions to enhance the rigour of our manuscript, which will make it stronger! Here "the first" means we are the first visual "large" language model that models temporal images, in contrast to the existing work (including ones you mentioned **[a,b,c,d]**) that are small transformer encoder-decoder models and their variants. To avoid confusion on this point, we will replace it with a rigorous statement (in **Section 1, line 91**):
>
> > To "We present Libra, the temporal-aware multimodal large language model (MLLM) capable of capturing temporal changes and overcoming the challenge of handling prior study citations, setting a new state-of-the-art performance in RRG tasks on the MIMIC-CXR dataset among MLLMs of the same parameter scale."
>
> Among the works you mentioned [a,b,c,d], we acknowledge that they address similar problems and utilise temporal scans but differ significantly in their methodologies. **[a]** employs symbolic alignment in its Longitudinal Projection Module and a separately trained BERT-based text generator. **RECAP [b]** uses a transformer encoder-decoder with symbolic task differentiation and two-stage training, starting with classification followed by report generation. **TiBiX [c]** adopts a transformer model with causal attention layers and learnable padding tokens to handle missing prior images. **BioViL-T [d]** is a self-supervised vision-language processing training framework that includes a CNN–Transformer hybrid multi-image encoder trained jointly with a BERT-based text model.
>
> * Since LLM-based models generally outperform other types of models in the RRG (Radiology Report Generation) task, the papers on **[a,b,c,d]** also do **not** compare their methods with LLM-based approaches.
>
> RECAP [b], TiBiX [c], and BioViL-T [d] were trained and tested on **MIMIC-CXR** [f], strictly adhering to the official dataset split. This is consistent with our experimental setup (as noted in **Section 3.1, line 265**), allowing for comparisons with Libra.
>
> However, model **[a]** was trained and tested on the **Chest ImaGenome** [e] dataset, which was automatically constructed from the MIMIC-CXR dataset. While the Chest ImaGenome dataset originates from MIMIC-CXR, it uses a **different train/valid/test split** and distinguishes between gold (manually validated and corrected) and silver (automatically generated) labels. The Chest ImaGenome dataset contains sentence-anatomy pair annotations, which are essential for training the visual anatomical token extraction module and performing sentence-anatomy dropout. This distinction is explicitly mentioned in [a]. Therefore, reproducing their model on the official MIMIC-CXR split for comparison with other models would be **inappropriate**.
>
> * In contrast, our approach integrates domain-specific knowledge within an **LLM** framework, leveraging pre-existing knowledge to learn more effectively without requiring task-specific modules. Meanwhile, the end-to-end manner is more effective than training from scratch, as done in [a,b,c,d].
>
> We sincerely appreciate your kind suggestion. In the revised manuscript, we will include a section providing a detailed discussion of both **non-LLM-based** and **LLM-based** models. This will help readers more comprehensively evaluate the performance of our model on the RRG task. The specific models discussed will include, but are not limited to, those you mentioned.
>
> *[a] Serra, Francesco Dalla, et al. "Controllable chest x-ray report generation from longitudinal representations."*
>
> *[b] Hou, Wenjun, et al. "RECAP: Towards Precise Radiology Report Generation via Dynamic Disease Progression Reasoning."*
>
> *[c] Sanjeev, Santosh, et al. "TiBiX: Leveraging Temporal Information for Bidirectional X-Ray and Report Generation."*
>
> *[d] Bannur, Shruthi, et al. "Learning to exploit temporal structure for biomedical vision-language processing."*
>
> *[e] Wu, Joy T., et al. "Chest imagenome dataset for clinical reasoning."*
>
> *[f] Johnson, Alistair EW, et al. "MIMIC-CXR, a de-identified publicly available database of chest radiographs with free-text reports."*

---

> > ### Comment · Reviewer_poux · 2024-11-26
> > **Response to authors' response**
> >
> > Thanks for trying to address the comment I raised. Most of these comments have been addressed in the response and some were promised to be considered should the paper be accepted. I have no more comments.

---

> > > ### Author Response · Authors · 2024-11-26
> > >
> > > Dear Reviewer poux,
> > >
> > >
> > > We are deeply grateful for your consideration of our responses and for raising your score. Your feedback has been invaluable in refining our work.
> > >
> > > As promised in the discussion, we have uploaded the revised manuscript with the updates incorporated. If you have any additional suggestions or concerns, we would be more than happy to address them to further enhance the quality of our manuscript.
> > >
> > > Thank you once again for your support and encouragement.
> > >
> > > Best regards,
> > >
> > > Authors

---

> ### Author Response · Authors · 2024-11-20
> **Response to Reviewer poux (2/3)**
>
> ***
> **W2. It is not clear whether the method is capable of handling one previous scan or multiple**
>
> **A2.**
>
> Our model processes only one previous scan. We will emphasise this point in the main content to reduce ambiguity.
>
> As clarified in **Appendix A.2 (line 890)**, we select the closest prior image as the single reference based on the official timestamps. Prior images are strictly retrieved from within the same data split to maintain data integrity, ensuring no data leakage occurs among the train/valid/test sets. **Section 2** provides a detailed description of our methodology, explaining how the model processes both the current image and a single prior/dummy image. In **line 184**, we clarify that if no prior image exists, the current image is used as a dummy prior image. Rather than handling the absence of a prior image in a symbolic way, our model leverages the specific characteristics of cases where a prior image is missing. This approach aligns with clinical practice, where temporal information exists between scans taken at two different time points, allowing the model to be trained in an end-to-end manner.
>
> * For example, if a patient has two scans just milliseconds apart, the current and prior images would be nearly identical, as no pathological changes would manifest within such **a short interval**. This extreme scenario simulates how the model handles clinical practice under limited temporal information.
> * In such case, the correct diagnosis for this tiny time interval would be that the patient's condition is **"stable"**. Thus, the dummy prior image addresses scenarios where a true prior image is unavailable. This approach improves the model's robustness by training it to correctly interpret temporal information even in cases with very short time intervals.
>
> This setup ensures the model consistently processes two image inputs (with a **single** previous scan) during both training and inference, with the current image as the primary input and the prior image as the auxiliary input. This design is reflected in the residual connections, as shown in **Figure 1**.
>
> Notably, our study focuses on **the scan** rather than on the **individual patient**. Therefore, our model is primarily designed to handle temporal information between two scans taken at different time points; processing more than two prior images is beyond the scope of this study. However, the interesting challenge of handling multiple prior images will be discussed in the **Limitations and Future Work** section.

---

> ### Author Response · Authors · 2024-11-20
> **Response to Reviewer poux (3/3)**
>
> ***
> **W3. In Figure 1, the effect of Rad-Dino is not clear. What if it is replaced with a classical pre-trained image encoder?**
> > **Review**: In Figure 1, the effect of Rad-Dino is not clear.
>
> **A3 (Answer-1).**
>
> We acknowledge that using specific image encoder names in **Figure 1** has caused some misunderstanding. We will replace them with the term "image encoder."
>
> In **Section 2.2 (line 107)**,  we mentioned that RAD-DINO [h] is continually pre-trained with medical scans by adopting the DINOv2 [i]. DINOv2 is an image-only self-supervised learning model, contrasting with image encoder like CLIP [j], which uses contrastive learning with both text and images.
>
> **RAD-DINO [h]** is a more advanced and domain-specific image encoder that has been widely used in the RRG task, including in works such as MAIRA-2 [k], M4CXR [l] and [h]. Therefore, we omit the details of RAD-DINO in the current manuscript. However, we will include a more detailed description of RAD-DINO as the appendix of the final version of the paper to make it more self-contained.
>
>
> > **Review**: What if it is replaced with a classical pre-trained image encoder?
>
> **A3 (Answer-2).**
>
> We indeed compared the effects of several different image encoders (**RAD-DINO**, **DINOv2** and **CLIP**), and the results demonstrate that RAD-DINO achieved better performance.
>
> * **Appendix B.1** details the effect of replacing the image encoder with a general-domain pre-trained model, specifically DINOv2. This further underscores the importance of domain-specific knowledge for the image encoder.
> * In **Appendix B.2**, we initialised a model with the CLIP image encoder to compare the effects of CLIP and DINO on the model.
>
> For other classical models, such as ResNet50-based image encoders, their performance on RRG tasks is limited, as validated in **[h]**. Hence, a detailed comparison of different types of pre-trained image encoders is currently outside the scope of our study.
>
> Meanwhile, in **line 68**, we mention that current MLLMs generally rely on a single hidden state layer from the pre-trained image encoder. In contrast, our model extracts features from all hidden layers and fuses them through the Temporal Alignment Connector, enabling the integration of image features with temporal information while aligning them with the dimensions of the LLM. Therefore, our study **focuses on** leveraging and aligning temporal images within MLLMs on the RRG task.
>
> Thanks for your constructive suggestion regarding image encoders. We would like to ask if you could share specific image encoders, as they were not mentioned in the review. In the revised manuscript, we will include a detailed discussion of various pre-trained encoders to make our work more comprehensive.
>
> *[h] Pérez-García, Fernando, et al. "RAD-DINO: Exploring Scalable Medical Image Encoders Beyond Text Supervision." arXiv preprint arXiv:2401.10815 (2024).*
>
> *[i] Oquab, Maxime, et al. "Dinov2: Learning robust visual features without supervision." arXiv preprint arXiv:2304.07193 (2023).*
>
> *[j] Radford, Alec, et al. "Learning transferable visual models from natural language supervision." International conference on machine learning. PMLR, 2021.*
>
> *[k] Bannur, Shruthi, et al. "Maira-2: Grounded radiology report generation." arXiv preprint arXiv:2406.04449 (2024).*
>
> *[l] Park, Jonggwon, et al. "M4CXR: Exploring Multi-task Potentials of Multi-modal Large Language Models for Chest X-ray Interpretation." arXiv preprint arXiv:2408.16213 (2024).*
>
> ***
> We sincerely hope that our responses have addressed your concerns and hope you will consider increasing your score. If we have left any notable points of concern overlooked, we would greatly appreciate your feedback, and we will attend to these points. Additionally, we will incorporate all the suggestions and discussions mentioned above in the latest manuscript. Thanks again for your thoughtful review and consideration.

---

### Comment · Area_Chair_fjLo · 2024-11-23
**Please review author response**

Dear reviewer,

Could you review the author response and let them know if you are satisfied with it or if you have any additional questions?

Kind regards,

Your AC

---

### Author Response · Authors · 2024-11-25
**Summary of main changes**

We sincerely thank all reviewers for their thoughtful and constructive feedback, which we believe will greatly enhance the clarity and quality of our work. We appreciate the thought-provoking questions and valuable suggestions for improving the manuscript. Below, we summarise the major changes, which we are confident have further strengthened the manuscript. All revisions in the text are marked in red, and newly added content is highlighted in blue.
## New experiments
* Ablation study for Temporal Alignment Connector(TAC) for stage 1, in Table 4. **(y9E4)**
* Ablation study for Temporal Alignment Connector(TAC) for stage 2, in Appendix C.3. **(y9E4)**
## Changes to the main text
* Replaced all instances of "visual language models (VLMs)" with "multimodal large language models (MLLMs)." **(poux)**
* The original ablation study about Temporal Alignment Connector (TAC) for stage 1, has been relocated to Appendix C.1 **(y9E4)**
* Clarify the formulas, in *Equation (1)* and *(7)*. **(Y2it)**
* Replaced the module colours, in Figure 1. **(Y2it)**

## Additional/Elaborated discussion

* Related work for LLM-based models and non-LLM-based models, in Appendix A.1. **(poux, Y2it)**
* Discussions on Radiological Image Representation from Image Encoders, in Appendix A.1. **(poux, Y2it)**
* Clarifications on Temporal Information and Research Objective, in Appendix A.2. **(poux, Y2it)**
* Limitations and future work, in Appendix A.3. **(ryyo)**
* Dataset Description and Selection, in Appendix B.2. **(y9E4)**
* Details of the Evaluation Metrics and Temporal Entity F1 Examples, in Appendix B.3. **(Y2it, y9E4)**
* Prompt Examples, in Appendix B.4. **(y9E4)**
* Comparison with Non-LLM-Based Models, in Appendix D.2. **(poux)**
* Detailed Heatmap Explanation about Figure 5, in Appendix E.1. **(y9E4)**
***
If you still need any clarification or have any other concerns, please feel free to contact us and we are happy to continue communicating with you.

Best regards,

Authors

---

> ### Author Response · Authors · 2024-11-26
>
> Dear Reviewers,
>
> We have updated the revised version of our manuscript. The new version includes additional experiments, analyses, theories, and explanations to address your concerns.
>
> If you have any further questions that have not been resolved, please do not hesitate to let us know. We would be delighted to continue communicating with you.
>
> Best regards,
>
> Authors

---

### Author Response · Authors · 2024-11-28
**Global Response**

Dear All Reviewers,

We would like to express our sincere gratitude for the valuable and constructive suggestions provided during the previous round of discussions. Your insights have been instrumental in refining and improving the revised manuscript.

We understand that some clarifications and explanations in our rebuttal may not have fully addressed all concerns. We acknowledge these gaps and remain committed to addressing them more effectively in future iterations of this work.

***
Given the extended timeline, we sincerely and humbly request that, if possible, you consider reassessing our revised manuscript. If the updates have sufficiently addressed your concerns, we would be deeply grateful if you would consider revisiting and potentially increasing your score.

If you believe there are still unresolved issues or specific areas where the manuscript could be further strengthened, we would greatly appreciate any additional suggestions. Your feedback continues to be invaluable in helping us achieve the highest possible standard for this work.

Thank you once again for your time, thoughtful feedback, and continued engagement.


Best regards,

Authors

---

### Meta-Review · Area_Chair_fjLo · 2024-12-22

**Metareview:**

This work proposes a temporal-aware framework for medical report generation that can take into account the previous scans to improve the quality of the generate reports. It is built upon the multimodal large language model, and the key component is the temporal alignment connector. Experimental study is conducted on benchmark datasets to demonstrate the efficiency of the proposed framework, and ablation study is provided to show the contribution of the key components.

Reviewers comment that this work studies an important problem, develops an innovation temporal processing module, and provides comprehensive evaluation. At the same time, the reviewers raised concerns related to the ability in handling multiple scans, the effectiveness of the temporal alignment connector, the clarification on experimental results, the framing of this work, and so on. The authors provide detailed answers to the raised concerns, which is appreciated. Reviewers poux ,Y2it, and y9E4 are overall satisfied with the rebuttal and Reviewer y9E4 increases the score to 6. At the same time, Reviewer ryyo clearly indicates that the clarification in the rebuttal is not particularly helpful or convincing and there is little new information presented in the rebuttal. During the further discussion between reviewers and AC, Reviewer ryyo highlights the two key concerns: 1) In reality, patients can have 1 or many scans, and this architecture does not elegantly handle any situation other than 2 images, and 2) The empirical evidence that the addition of a prior scan boosts performance via this architecture is very weak. Reviewer y9E4 also agrees that the response did not fully resolve the issues raised by Reviewer ryyo. The final ratings are 5, 5, 6, and 5.

AC carefully reviews the submission, the comments, the rebuttals, and the discussion. AC can see that this work has its merits in attempting to address an important issue in medical report generation. Meanwhile, AC also agrees with the reviewers on the raised concerns, especially the improvement brought by the proposed framework in considering the previous scan. It can be seen that the improvement is only significant for the F1_temp score proposed by this work while the improvement on other commonly used metrics is overall marginal. As a result, the real contribution and benefit of the proposed temporal alignment connector module becomes unclear. Taking all the factors into account, this work in its current form cannot be recommended for acceptance. It is hoped that the reviews could help to further improve the quality of this work.

**Additional Comments On Reviewer Discussion:**

The reviewers raised concerns related to the ability in handling multiple scans, the effectiveness of the temporal alignment connector, the clarification on experimental results, the framing of this work, and so on. The authors provide detailed answers to the raised concerns, which is appreciated. Reviewers poux ,Y2it, and y9E4 are overall satisfied with the rebuttal and Reviewer y9E4 increases the score to 6. At the same time, Reviewer ryyo clearly indicates that the clarification in the rebuttal is not particularly helpful or convincing and there is little new information presented in the rebuttal. During the further discussion between reviewers and AC, Reviewer ryyo highlights the two key concerns: 1) In reality, patients can have 1 or many scans, and this architecture does not elegantly handle any situation other than 2 images, and 2) The empirical evidence that the addition of a prior scan boosts performance via this architecture is very weak. Reviewer y9E4 also agrees that the response did not fully resolve the issues raised by Reviewer ryyo. The final ratings are 5, 5, 6, and 5.

AC carefully reviews the submission, the comments, the rebuttals, and the discussion. AC can see that this work has its merits in attempting to address an important issue in medical report generation. Meanwhile, AC also agrees with the reviewers on the raised concerns, especially the improvement brought by the proposed framework in considering the previous scan. It can be seen that the improvement is only significant for the F1_temp score proposed by this work while the improvement on other commonly used metrics is overall marginal. As a result, the real contribution and benefit of the proposed temporal alignment connector module becomes unclear. Taking all the factors into account, this work in its current form cannot be recommended for acceptance. It is hoped that the reviews could help to further improve the quality of this work.

---

### Decision · Program_Chairs · 2025-01-22

Reject